# SlitC-PlexinA1 mediates iterative inhibition for orderly passage of spinal commissural axons through the floor plate

Hugo Ducuing[1†], Thibault Gardette[1†], Aurora Pignata[1], Karine Kindbeiter[1], Muriel Bozon[1], Olivier Thoumine[2], Céline Delloye-Bourgeois[1], Servane Tauszig-Delamasure[1], Valerie Castellani[1]*

[1]Institut NeuroMyoGène - CNRS UMR 5310 - INSERM U1217 de Lyon- UCBL Lyon 1, Faculté de Médecine et de Pharmacie, Lyon, France; [2]Interdisciplinary Institute for Neuroscience, UMR CNRS 5297 - University of Bordeaux, Bordeaux, France

**Abstract** Spinal commissural axon navigation across the midline in the floor plate requires repulsive forces from local Slit repellents. The long-held view is that Slits push growth cones forward and prevent them from turning back once they became sensitized to these cues after midline crossing. We analyzed with fluorescent reporters Slits distribution and FP glia morphology. We observed clusters of Slit-N and Slit-C fragments decorating a complex architecture of glial basal process ramifications. We found that PC2 proprotein convertase activity contributes to this pattern of ligands. Next, we studied Slit-C acting via PlexinA1 receptor shared with another FP repellent, the Semaphorin3B, through generation of a mouse model baring PlexinA1$_{Y1815F}$ mutation abrogating SlitC but not Sema3B responsiveness, manipulations in the chicken embryo, and ex vivo live imaging. This revealed a guidance mechanism by which SlitC constantly limits growth cone exploration, imposing ordered and forward-directed progression through aligned corridors formed by FP basal ramifications.

**\*For correspondence:**
valerie.castellani@univ-lyon1.fr

[†]These authors contributed equally to this work

**Competing interests:** The authors declare that no competing interests exist.

## Introduction

During neural circuit formation, axons are thought to navigate series of intermediate targets or choice points until they reach their final destination (*Comer et al., 2019*; *Raper and Mason, 2010*). This concept emerged from seminal studies of limb bud innervation in insects (*Bate, 1976*; *Bentley and Keshishian, 1982*; *Ho and Goodman, 1982*; *Palka et al., 1992*) and has been generalized to all contexts of long-distance axon projections. First, intermediate targets attract the axons, and second, set their response to the 'next-step' guidance cues that will specify their upcoming trajectory at the exit (*Avilés and Stoeckli, 2016*; *Zuñiga and Stoeckli, 2017*). They also provide a variety of extracellular cues to guide the axons along the navigation (*Nawabi and Castellani, 2011*; *Neuhaus-Follini and Bashaw, 2015*). This makes the navigation of intermediate targets a critical and complex process where the activity of the different cues guiding the axons at each step must be coordinated. This is of particular importance for the trajectory changes elicited by the 'next-step' cues that must occur only after completion of the crossing. It has been proposed that this matching is achieved through spatio-temporal regulation of the sensitivity of axon terminals, the growth cones, that gain responsiveness to the 'next-step' guidance cues at the exit (*Avilés and Stoeckli, 2016*; *Zuñiga and Stoeckli, 2017*). Intriguingly, in various in vivo manipulations of guidance signaling, axons are observed to skip or incompletely achieve the intermediate target crossing, prematurely re-orienting their trajectory, which compromises the formation of neural circuits (*Zou et al., 2000*; *Long et al., 2004*; *Chen et al., 2008*; *Philipp et al., 2012*; *Delloye-Bourgeois et al., 2015*; *Yang et al., 2018*; *Bagri et al., 2002*; *Friocourt and Chédotal, 2017*). This suggests that the axons

can perceive the 'next-step' cues during or even prior to the intermediate target navigation. Thus possibly, some mechanisms exist that counteract the action of the 'next-step' cues until completion of intermediate target navigation.

The formation of spinal cord commissural circuits provides a very suitable model to address the mechanisms of intermediate target navigation. Commissural axons cross the midline in a key intermediate target located at the ventral side of the neural tube, the floor plate (FP). Glial cells, composing the FP, have a typical progenitor-like bipolar morphology, with a dorsal apical anchor at the lumen of the central canal and a basal process laying ventrally on the basal lamina, even though some studies suggested these cells have complex morphology (*Campbell and Peterson, 1993*). Commissural axons path between the glia soma and the basal lamina. Despite seminal electronic microscopy observations that reported close axon–glia contacts (*Yaginuma et al., 1991*), little is known about the exact morphology of FP basal processes. Slit1-3 and Semaphorin3B (Sema3B) cues produced by the FP glia were proposed to help the FP navigation. PlexinA1 (PlxnA1) is a commissural axon receptor shared by Slit and Semaphorin ligands. When associated with Neuropilin2 (Nrp2), it mediates the response to Semaphorin3B (Sema3B) (*Nawabi et al., 2010*), and while unbound to Nrp2, it can act as a receptor for the C-terminal fragment of Slit1-3 (SlitC) (*Delloye-Bourgeois et al., 2015*). SlitC is generated from the processing of integral Slit1-3 proteins together with Slit1-3-N terminal fragment, whose functions are mediated by Roundabout (Robo) receptors (*Evans and Bashaw, 2010*). Both the spatial distribution and mode of action of Slits and Sema3B proteins remain unclear but these cues are thought to push the axons toward the contralateral side and also to prevent any turning back (*Ducuing et al., 2019*). Nevertheless, commissural phenotypes resulting from genetic deletions of ligands or their receptors suggested that SlitC, SlitN, and Sema3B might bring specific contributions to commissural axon navigation (*Ducuing et al., 2019*; *Chédotal, 2019*). Lastly, at the FP exit, commissural axons accomplish a drastic longitudinal turn directed by two synergizing rostro-caudal gradients of Wnt and Sonic Hedgehog, to navigate rostrally toward the brain (*Avilés and Stoeckli, 2016*; *Zuñiga and Stoeckli, 2017*; *Lyuksyutova et al., 2003*; *Bourikas et al., 2005*). Various regulations activated during the FP navigation were reported to occur in commissural axons that switch on their sensitivity to the 'next-step' cues after completion of the crossing (*Yam et al., 2012*; *Onishi and Zou, 2017*; *Avilés et al., 2013*).

To gain novel insights into the process of FP navigation, we focused in the present study on the SlitC-PlxnA1 signaling. Through combinations of in vivo analysis, live imaging, and super resolution microscopy, we show that SlitC does not act by pushing commissural axons after crossing. Instead, it is deposited onto remarkably complex ramifications of FP glia basal process that mark out the entire FP navigation path. Reiterated contacts with ligand-coated glial surfaces produce continuous constrains that protect the growth cones from the risk of aberrant deviations that could result from active exploration and response to other guidance cues of their local environment.

## Results

### PlxnA1$_{Y1815F}$ mutation in mice recapitulates PlxnA1$_{-/-}$ recrossing but not stalling phenotype

We identified in previous work a tyrosine residue of PlxnA1 cytoplasmic tail, whose phosphorylation was specifically triggered upon SlitC exposure (*Delloye-Bourgeois et al., 2015*). Expression of PlxnA1$_{Y1815F}$ mutant receptor in commissural neurons significantly reduced their SlitC but not Sema3B induced growth cone collapse response (*Delloye-Bourgeois et al., 2015*). Re-crossing phenotype in embryos lacking PlxnA1 or all Slit1-3 but not Robo1/2 or Sema3B (*Delloye-Bourgeois et al., 2015*) suggested that PlxnA1-SlitC signaling prevents back crossing. To test this model, we generated a PlxnA1$_{Y1815F}$ mouse strain (*Figure 1—figure supplement 1*), and validated PlxnA1 expression in western blot on spinal cord lysates (*Figure 1—figure supplement 1*). We analyzed the commissural tracts using the lipophilic dye I (DiI) as in *Delloye-Bourgeois et al., 2015*. Interestingly, while no difference of stalling was observed between PlxnA1$_{+/+}$ and PlxnA1$_{Y1815F/Y1815F}$ embryos, the recrossing cases were exclusively associated with loss of one or two PlxnA1 gene copies, recapitulating PlxnA1$_{-/-}$ embryos phenotype (*Figure 1A and B*). Moreover, in the $_{Y1815F}$ context, axon trajectories appeared disorganized, with axons baring a tortuous aspect, even when they could achieve FP crossing (*Figure 1A*).

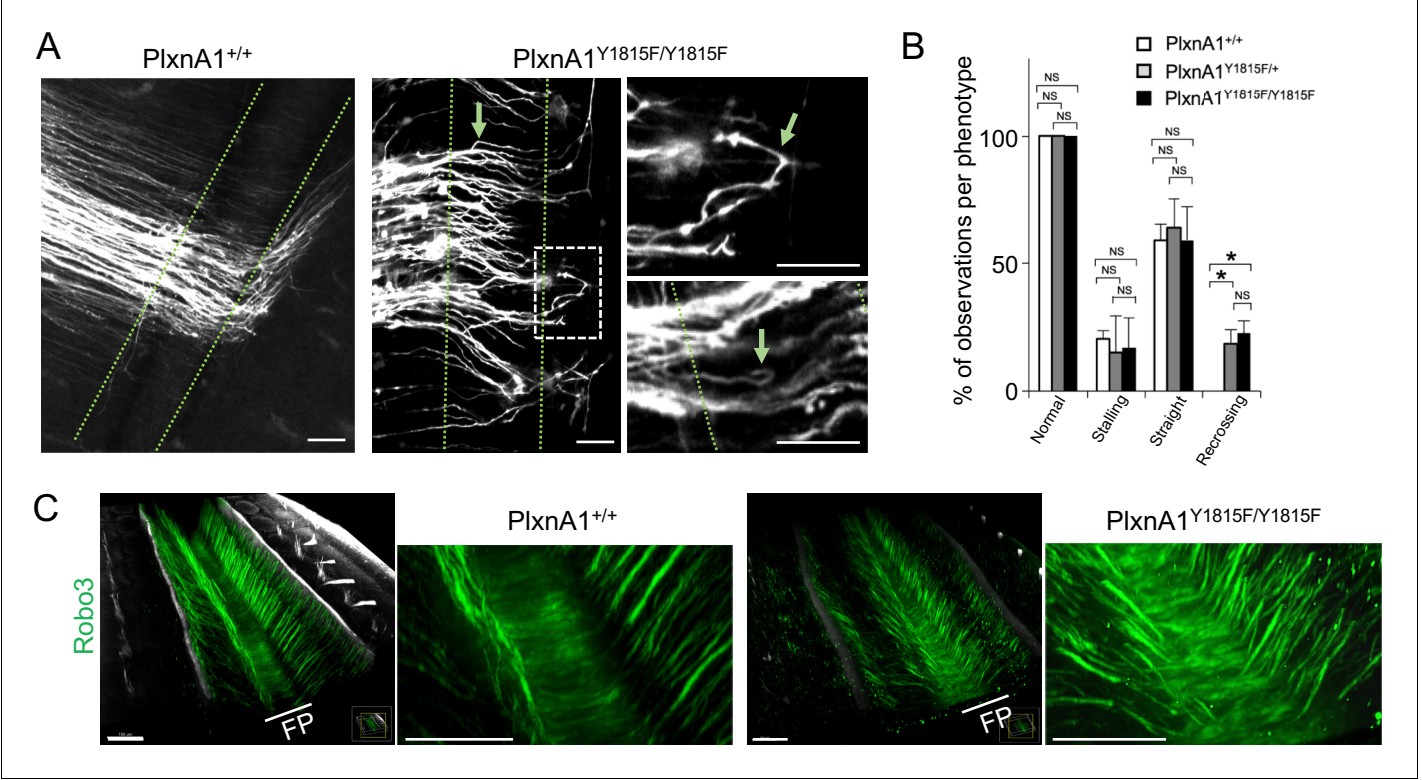

**Figure 1.** Y1815F mutation in PlxnA1 induces commissural axon recrossing and disorganized trajectories. (**A**) Microphotographs illustrating commissural tracts of PlxnA1$_{+/+}$ and PlxnA1$_{Y1815F/Y1815F}$ E12.5 open-books labeled with DiI. In PlxnA1$_{+/+}$ embryos, axons extend straight toward the floor plate (FP), cross the FP, and turn rostrally at the FP exit. In PlxnA1$_{Y1815F/Y1815F}$ embryos, some axons turn back or are misdirected during the navigation of the FP (indicated by green arrows). The FP is delimited by dashed green lines. (**B**) Quantitative analysis of commissural axon phenotypes (PlxnA1$_{+/+}$, N = 3 embryos; PlxnA1$_{+/Y1815F}$, N = 5 embryos; PlxnA1$_{Y1815F/Y1815F}$, N = 4 embryos). Data are shown as the mean ± s.e.m., Student's t-test has been applied, *: p<0.05. (**C**) Light sheet imaging of the spinal commissural tracts in PlxnA1$_{+/+}$ and PlxnA1$_{Y1815F/Y1815F}$ embryos at E12.5, immunostained with anti-Robo3 antibody. Scale bar: 50 μm in (**A**), 150 μm in (**C**).

The online version of this article includes the following source data and figure supplement(s) for figure 1:

**Source data 1.** Quantitative analysis of commissural axon phenotypes (B).

**Figure supplement 1.** Generation of PlxnA1$_{Y1815F}$ mutant strain.

**Figure supplement 1—source data 1.** Percentage of mice with each genotype coming from PlxnA1$_{Y1815F/+}$ × PlxnA1$_{Y1815F/+}$ crossing (C).

**Figure supplement 1—source data 2.** Overall percentage of female and male mice (D).

**Figure supplement 2.** DiI traced commissural axon trajectories in PlxnA1$_{Y1815F}$ embryos are disorganized in the floor plate (FP) but not obviously prior to the crossing.

Thus, the ability of commissural axons to maintain a straight trajectory toward the FP exit requires a functional PlxnA1-SlitC signaling. In contrast, the lack of stalling phenotype in PlxnA1$_{Y1815F/Y1815F}$ embryos indicates that the guidance forces resulting from the unaffected PlxnA1-Nrp2-Sema3B signaling remain sufficient for pushing commissural axons toward the FP exit.

## PlxnA1$_{Y1815F}$ mutation does not obviously affect pre-crossing commissural navigation

Next, transverse sections of E12.5 embryos were immunostained with antibodies directed against pre-crossing Robo3 and post-crossing L1CAM commissural markers. Their pattern revealed no obvious defects of coursing toward the FP, indicating that PlxnA1 Y1815 residue is dispensable for the pre-crossing navigation (*Figure 1—figure supplement 2*). This is consistent with our previous finding that PlxnA1 is not delivered at the growth cone surface prior to the FP entry, thus having no expected pre-crossing role (*Pignata et al., 2019*). PlxnA1 expression was also not obviously different between WT and PlxnA1$_{Y1815F/Y1815F}$ embryos, detected in both cases at highest levels in crossing and post-crossing axons (*Figure 1—figure supplement 2*).

Thus, the Y1815 mutation does not prevent PlxnA1 synthesis nor its trafficking to the axon.

## PlxnA1$_{Y1815F}$ expressing axons fail to maintain forward directed growth when navigating the FP

To gain further insights, we first reintroduced WT and mutated PlxnA1 in E12.5 PlxnA1$_{-/-}$ murine embryos by ex vivo electroporation. We constructed a pHluo tagged version of PlxnA1$_{Y1815F}$ receptor, allowing reporting the receptor at the cell membrane (*Pignata et al., 2019*). The pHluo-PlxnA1$_{Y1815F}$ and pHluo-PlxnA1$_{WT}$ constructs were co-electroporated along with a mbTomato construct (*Figure 2A*). Right after electroporation, the spinal cords were dissected out and open-books were incubated for 2 days and imaged at fixed time-point with a spinning disk confocal microscope. We analyzed two different aberrant behaviors: axons strongly deviating from their normal straight trajectory (>30°), and axons that accomplished premature turning within the FP or turning back. Interestingly, the proportion of straight trajectory decreased from 86% in the WT condition to 54% in the PlxnA1$_{Y1815F}$ one. Conversely, we observed significant increase of deviations (from 9% to 20%) and premature turns (from 5% to 27%) for pHluo-PlxnA1$_{Y1815F+}$ axons compared to pHluo-PlxnA1$_{WT+}$ ones (*Figure 2B and D*). We also counted the number of deviations of individual axons that succeeded navigating the FP, from 0 (straight or slightly curved at a low angle) up to more than two deviation angles. Interestingly, pHluo-PlxnA1$_{WT+}$ growth cones exhibited 73% of straight trajectories while PlxnA1$_{Y1815F}$ ones only 36%. The latter displayed much more irregular curly aspects, with trajectories having significantly increased number of curvatures (*Figure 2C and D*). Thus, and consistent with our analysis of PlxnA1$_{Y1815F/Y1815F}$ mouse model, re-expressing a PlxnA1 receptor deficient for SlitC signaling in PlexnA1$_{-/-}$ open-books resulted in strong alteration of the FP navigation.

Video-time lapse with this paradigm turned out to be highly challenging, mainly due to high and fast toxicity resulting from repeated laser exposure. Hence, we turned to the chicken embryo as an alternative model. We electroporated pHluo-PlxnA1$_{WT}$ and pHluo-PlxnA1$_{Y1815F}$ in the neural tube of chicken embryos for time lapse analysis of the commissural axon navigation as in *Pignata et al., 2019*. We plotted the trajectories of individual axons at successive time points, classifying them as straight, tortuous, and premature turning/turning back. We also assessed growth cone stalling. We found that 63% of axons had a straight trajectory in the PlxnA1$_{WT}$ condition, for only 34% in the PlxnA1$_{Y1815F}$ condition. Conversely, 36% of PlxnA1$_{Y1815F}$ axons were tortuous, for only 17% for the PlxnA1$_{WT}$ ones. Moreover, the proportion of early turns increased from 11% to 19% in the PlxnA1$_{Y1815F}$ condition compared to the PlxnA1$_{WT}$ one (*Figure 2E–G*). The rate of growth cone stalling was comparable in both conditions (9% and 10% for PlxnA1$_{WT}$ and PlxnA1$_{Y1815F}$ respectively), which was also consistent with the lack of increased stalling phenotype in PlxnA1$_{Y1815F/Y1815F}$ mouse embryos. We also analyzed commissural axon trajectories to quantify their degree of deviation. As in mouse open-books, we found the percentage of straight trajectories significantly decreased from 83% in the PlxnA1$_{WT}$ condition to 53% in the PlxnA1$_{Y1815F}$ one, with correlated increase in curved shapes (*Figure 2H–J*).

Altogether these experiments confirmed that PlxnA1$_{Y1815F}$ is unable to ensure the forward growth direction that is normally taken by commissural axons to cross the FP and validated the use of the chicken embryo as a model for further investigations.

## The temporal and spatial pattern of PlxnA1 membrane insertion during commissural axon navigation is impacted by the Y1815 mutation

Next, we investigated in living conditions whether the Y1815 mutation alters SlitC-PlxnA1 signaling by modifying PlxnA1 dynamics during the FP navigation. First, we analyzed the pattern of pHluo receptor introduced in PlxnA1$_{-/-}$ mouse embryos. As expected from PlxnA1 expression in PlxnA1$_{Y1815F}$ transverse sections, pHluo-PlxnA1$_{Y1815F}$ was trafficked to the axon and the growth cones. However, while PlxnA1$_{WT}$ was mostly restricted to the growth cone, PlxnA1$_{Y1815F}$ occupied a much larger membrane domain, overflowing in the adjacent axon shaft compartment (*Figure 3A and B*). Thus, the Y1815 mutation alters PlxnA1 cell surface pattern in navigating commissural axons.

Second, because the endogenous PlxnA1 receptor is still present in the chicken embryo model, we wondered whether the dynamics of PlxnA1$_{WT}$ is altered by the expression of PlxnA1$_{Y1815F}$. To address this question, we set a competition assay by electroporating pHluo-PxnA1$_{WT}$ either with

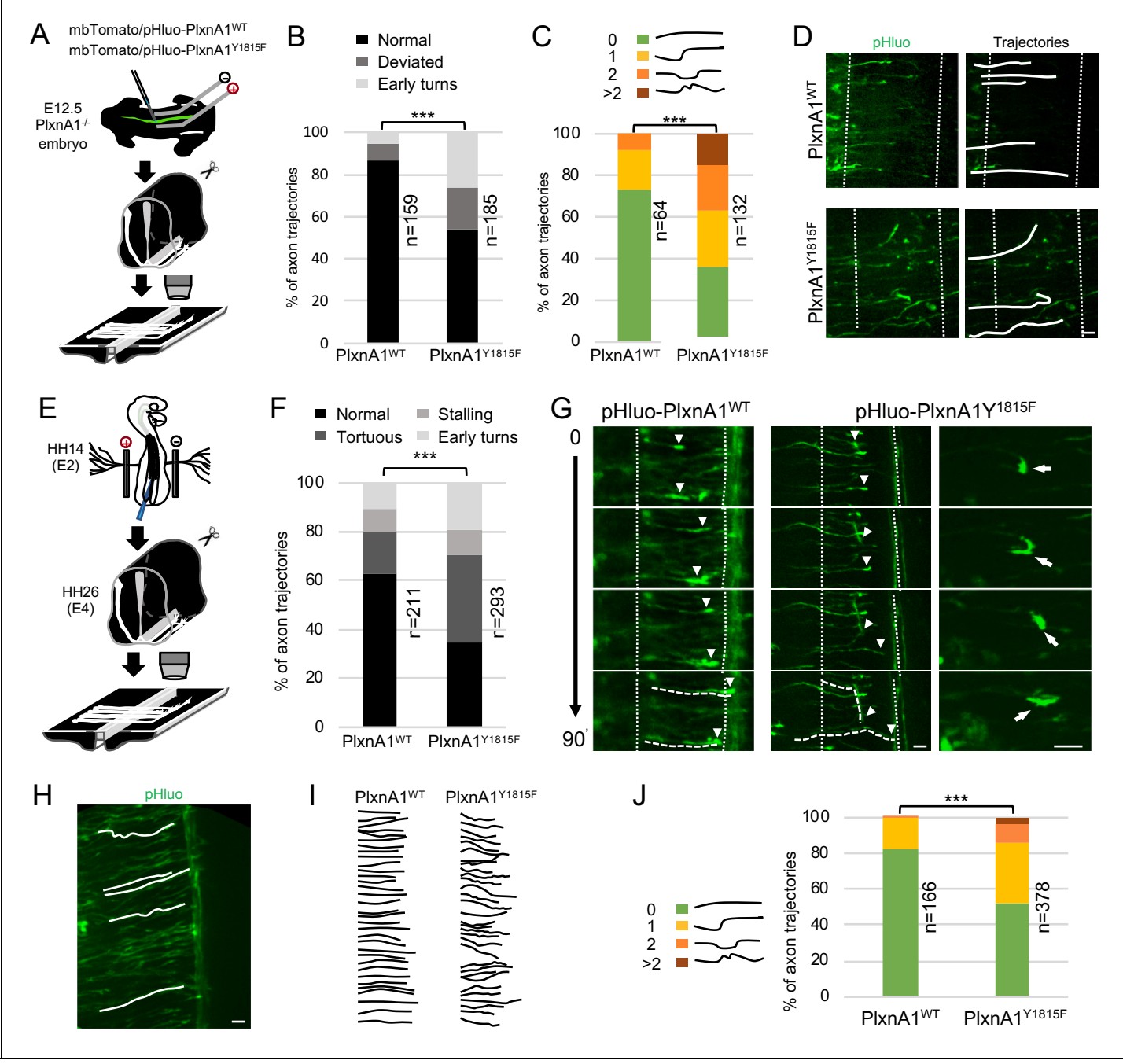

**Figure 2.** Commissural neurons expressing PlxnA1_{Y1815F} at their surface fail to maintain straight growth during floor plate (FP) navigation. (**A**) Schematic drawing of the paradigm of electroporation of pHluo-PlxnA1 receptor forms in PlxnA1_{-/-} mouse embryonic spinal cord and open-book preparations for live imaging. (**B**) Histogram of axon trajectory phenotypes (PlxnA1_{WT}, N = 8 embryos, 159 growth cones; PlxnA1_{Y1815F}, N = 7 embryos, 185 growth cones). Chi-squared test has been applied, ***: p<0.001. (**C**) Histogram depicting the analysis of axon trajectories, classified by counting curvatures according to the indicated criteria (PlxnA1_{WT}, N = 10 embryos, 64 growth cones; PlxnA1_{Y1815F}, N = 7 embryos, 132 growth cones). (**D**) Microphotographs illustrating navigating commissural growth cones expressing the WT and the mutated pHluo-receptor at their surface, reported by the green pHluo fluorescent signal. (**E**) pHluo-receptor constructs were co-electroporated with the mbTomato in the chicken neural tube and spinal cords were mounted in open-books for time-lapse imaging. (**F**) Histogram quantifying axon trajectories reconstructed from time-lapse sequences of pHluo+ growth cones navigating the FP (PlxnA1_{WT}, N = 3 embryos, 211 growth cones; PlxnA1_{Y1815F}, N = 3 embryos, 293 growth cones). (**G**) Snapshots of the movies illustrating the commissural growth patterns (white triangles) and an illustration of a PlxnA1_{Y1815F+} growth cone turning back. The FP is delimited by dashed white lines. (**H**) Microphotograph illustrating traces of commissural axon trajectories. (**I**) Example of traces patterns from several snapshots. (**J**) Histogram depicting the analysis of axon trajectories, classified by counting curvatures according to the indicated criteria (PlxnA1_{WT},

*Figure 2 continued on next page*

*Figure 2 continued*

N = 3 embryos, 166 growth cones; PlxnA1$_{Y1815F}$, N = 3 embryos, 378 growth cones). In (**C, F, and J**) Chi-squared test has been applied, \*\*\*: p<0.001.
Scale bars: 10 μm in (**D, G, and H**).
The online version of this article includes the following source data for figure 2:

**Source data 1.** Histogram of axon trajectory phenotypes (B).
**Source data 2.** Histogram depicting the analysis of axon trajectories, classified by counting curvatures according to the indicated criteria (C).
**Source data 3.** Histogram quantifying axon trajectories reconstructed from time-lapse sequences of pHluo+ growth cones navigating the FP (F).
**Source data 4.** Histogram depicting the analysis of axon trajectories, classified by counting curvatures according to the indicated criteria (J).

non-fluorescent his-PlxnA1$_{WT}$ or with flag-PlxnA1$_{Y1815F}$. We then recorded in open-books the surface distribution of pHluo-PlxnA1$^{WT}$ in commissural axons navigating the FP. We found in both electroporation conditions that pHluo-PlxnA1$^{WT}$ surface pool was enriched in the growth cones, compared to adjacent axon shaft, as observed in our experiments with electroporated PlxnA1$^{-/-}$ mouse open-books (*Figure 3—figure supplement 1*). No statistical difference was found between the two conditions. Moreover, the cell surface expression levels of pHluo-PlxnA1$^{WT}$ were not significantly altered by co-expression with PlxnA1$_{Y1815F}$ suggesting no indirect impact of PlxnA1$_{Y1815F}$ on the WT receptor (*Figure 3—figure supplement 1*).

Third, we recently reported using chick open-books that PlxnA1 is specifically delivered at the growth cone surface when commissural axons navigate the first FP half, while Robo1 is sorted in the second half (*Pignata et al., 2019*). We studied pHluo-PlxnA1$_{Y1815F}$ dynamics in this paradigm by video-microscopy, plotting the position of growth cones switching on the pHluo fluorescence. Interestingly, in both PlxnA1$_{WT}$ and PlxnA1$_{Y1815F}$ conditions, nearly 100% of the growth cones navigating the FP had sorted the receptor before midline crossing, indicating that the temporal pattern of membrane insertion within the FP was unaffected by the Y1815 mutation (*Figure 3C and D* and *Videos 1* and *2*). Nevertheless, we also observed that a significant proportion of PlxnA1$_{Y1815F+}$ growth cones, which could navigate the FP, had sorted the receptor prior to the FP entry (*Figure 3C* and *Videos 3* and *4*).

This suggested that the mutated receptor might be sorted precociously, although it appeared not to arrest the growth cones at the FP entry.

Fourth, we recently reported using Atto-647N-conjugated green fluorescent protein (GFP) nanobodies and STED microscopy that the membrane pool of PlxnA1$_{WT}$ concentrates at the front of growth cones navigating the FP (*Pignata et al., 2019*). PlxnA1$_{Y1815F}$ distribution was studied in parallel, using the same experimental conditions. Notably, in sharp contrast with our observations of PlxnA1$_{WT}$, PlxnA1$_{Y1815F}$ spanned from the front to the rear of the growth cone (*Figure 3E–G*).

Thus, PlxnA1$_{Y1815F}$ spatial distribution in commissural axons navigating the FP is altered, which reveals that the mutation might interfere with SlitC-specific traffic and signaling.

## Y1815F results in increased membrane mobility of PlxnA1 receptor in navigating commissural growth cones

Next, we conducted fluorescence recovery after photobleaching (FRAP) experiments to compare exocytosis and membrane motility of WT and mutated PlxnA1 in commissural growth cones navigating the FP. The pHluo-receptor fluorescence was bleached, and then the fluorescence recovery was recorded for 20 min (*Figure 3H–K* and *Videos 5–8*). Strikingly, in the PlxnA1$_{Y1815F}$ condition, the fluorescence recovery was significantly higher over the first time points, than in the PlxnA1$_{WT}$ one, and the difference established from this early step remained constant over time. Thus, PlxnA1$_{Y1815F}$ has a faster membrane diffusion than PlxnA1$_{WT}$. The stability of the difference over time also suggests that exocytosis events might be unaffected by the mutation. Thus, Y1815F mutation results in relapse of PlxnA1 mobility at the cell membrane which, either upstream or downstream of SlitC binding, could affect the ability of the receptor to mediate SlitC signal.

## FP glia cells elaborate complex ramified basal processes staking the axon path

Next, to gain insights into how SlitC maintains a straight growth of commissural axons, we wanted to get a precise analysis of the topology of the axon navigation path in the FP and to characterize

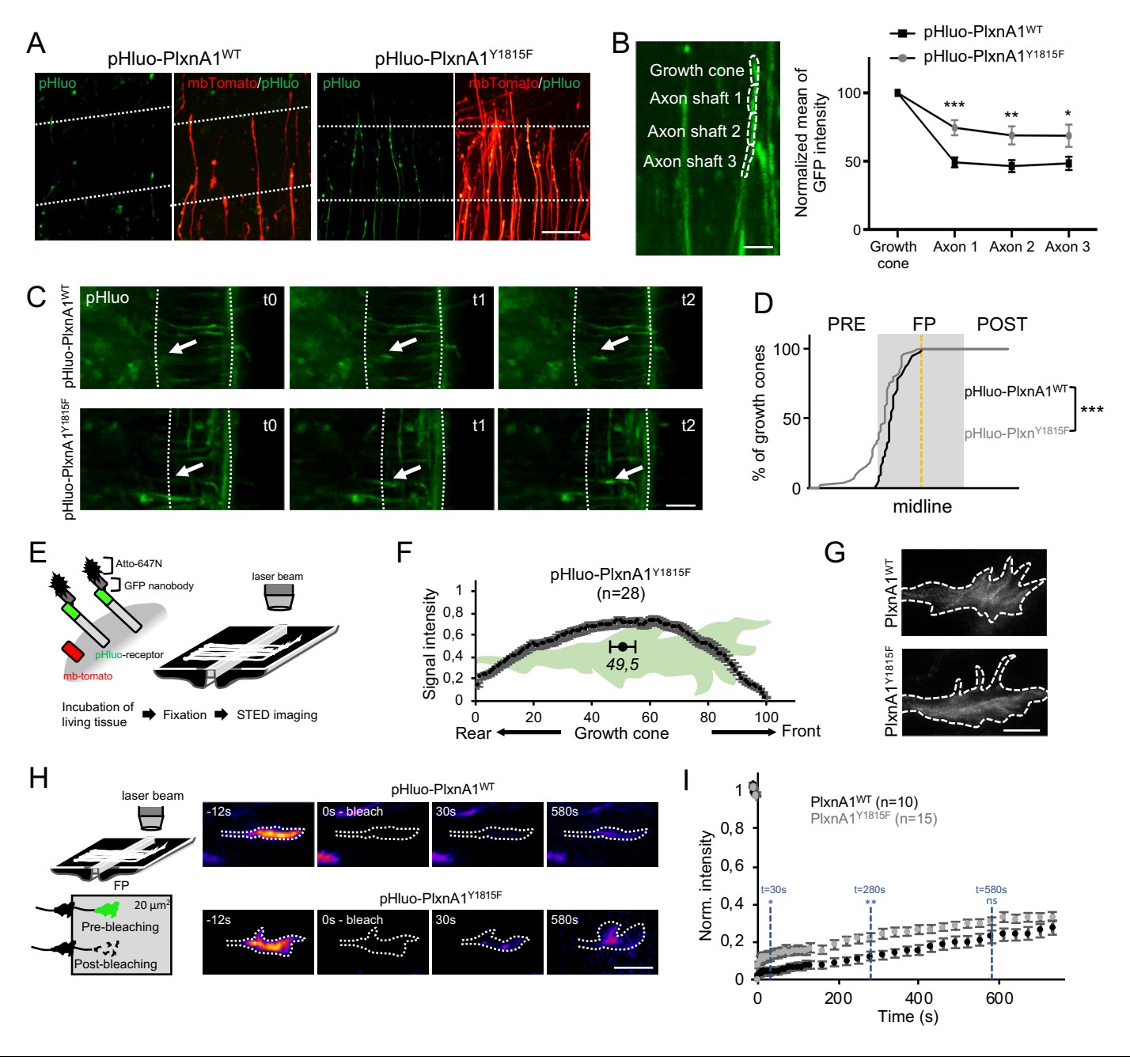

**Figure 3.** The Y1815F mutation alters the spatio-temporal pattern of cell surface PlxnA1 distribution and membrane mobility at the growth cone. (**A**) Microphotographs of commissural axons navigating the floor plate (FP) in living open-books of electroporated PlxnA1-/- mice embryos. The FP is delimited by dashed white lines. (**B**) Method of quantification and quantification of the pHluo signal (PlxnA1$_{WT}$, N = 6 embryos, 24 growth cones; PlxnA1$_{Y1815F}$, N = 4 embryos, 18 growth cones). Data are shown as the mean ± s.e.m., Student's t-test has been applied, *: p<0.05, **: p<0.01, ***: p<0.001. (**C**) Microphotographs of live imaging movies illustrating the sorting of PlxnA1 receptor at the surface of commissural growth cones navigating the FP (delimited by dashed white lines). (**D**) Cumulative fractions of growth cones that sort PlxnA1 to their surface, reported by the pHluo fluorescence, in electroporated chick embryos (pHluo-PlxnA1$_{WT}$, N = 2 embryos, 38 growth cones; pHluo-PlxnA1$_{Y1815F}$, four embryos, 53 growth cones). KS test has been applied, ***: p<0.001. (**E**) Schematic representation of the paradigm of STED imaging in open-books. (**F**) Quantification of the center of mass of the signal (N = 28 growth cones from five embryos). Data are shown as the mean ± s.e.m. (**G**) STED microscopy images of pHluo-PlxnA1$_{Y1815F}$ cell surface distribution in commissural growth cones navigating the FP. (**H**) Schematic representation of the paradigm of fluorescence recovery after photobleaching (FRAP) experiments. Representative color-coded images from video-time lapse sequences illustrating photobleaching and fluorescence recovery. (**I**) Graphs of fluorescence recovery for pHluo-PlxnA1$_{WT}$ and pHluo-PlxnA1$_{Y1815F}$ (PlxnA1$_{WT}$, N = 2 embryos, 10 growth cones; PlxnA1$_{Y1815F}$,

*Figure 3 continued on next page*

*Figure 3 continued*

N = 2 embryos, 15 growth cones). Data are shown as the mean ± s.e.m., Student's t-test has been applied at t = 30 s, t = 280 s, and t = 580 s, ns: non-significant, *: p<0.05, **: p<0.01. Scale bars: 5 µm in (**G**), 10 µm in (**B and H**), 50 µm in (**A and C**).

The online version of this article includes the following source data and figure supplement(s) for figure 3:

**Source data 1.** Quantification of the pHluo signal (B).
**Source data 2.** Cumulative fractions of growth cones that sort PlxnA1 to their surface, reported by the pHluo fluorescence, in electroporated chick embryos (D).
**Source data 3.** Quantification of the center of mass of the signal (F).
**Source data 4.** Graphs of fluorescence recovery for pHluo-PlxnA1$_{WT}$ and pHluo-PlxnA1$_{Y1815F}$ (I).
**Figure supplement 1.** pHluo-PlxnA1$_{WT}$ has similar distribution pattern in commissural axons navigating the floor plate (FP) when co-expressed with non-fluorescent PlxnA1WT or PlxnA1$_{Y1815F}$ in chick open-books.
**Figure supplement 1—source data 1.** Histogram of the quantification of the pHluo signal (B).
**Figure supplement 1—source data 2.** Quantification of the pHluo signal in growth cones and proximal shafts of commissural axons navigating the FP (C).

the spatial distribution of the different ligands (*Figure 4A*). We first immunostained transverse sections of E4 chick embryos with an antibody recognizing the FP glial cell marker Ben. The staining revealed a dense network of processes in the basal compartment corresponding to the axon path, in particular with BEN+ structures aligned in the left–right axis (*Figure 4B*). This was consistent with the architecture of basal process spanning the navigation path reported in previous work, supporting that commissural growth cones establish contacts with glia cell surfaces during FP crossing (*Campbell and Peterson, 1993*; *Yaginuma et al., 1991*; *Okabe et al., 2004*).

We next performed a sparse electroporation of mbTomato in chick embryos to examine individual cells and their arrangement in the three dimensions of the tissue. We observed that the basal process of the glial cell is not a simple extension but rather is a ramified structure (*Figure 4D and E*). 3D reconstruction with IMARIS software after deconvolution treatment of the images revealed a 'squid-like' structure, with a basal process subdivided into several pillars, with typically several anchors to the ventral pole (*Figure 4F*). The pillars appeared much larger in the left–right axis than in the rostro-caudal one, as suggested by previous data from electronic microscopy (*Okabe et al., 2004*; *Figure 4E and F*). Next, we electroporated mbTomato under a Math1 promoter, which specifically drives the expression in dorsal commissural neurons (*Helms and Johnson, 1998*) and immunostained the electroporated samples with an anti-BEN antibody. We observed that commissural growth cones are individually intercalated between glial cell processes (*Figure 4G*). We finally co-electroporated Math1-mbTomato with HoxA1-GFP specifically driving the GFP in FP cells (*Li and Lufkin, 2000*; *Figure 4H and I*). From confocal images of Math1-mbTomato fluorescent signal, we reconstructed the morphology of commissural growth cones. We observed them infiltrating the space between pillars, having an oblong shape, flattened in the left–right dimension and elongated in the dorso-ventral one (*Figure 4J*). Thus, FP glia elaborate highly complex mesh of basal processes whose stereotypic spatial organization provides a physical frame for commissural growth cones, likely imposing numerous and repeated contacts all over the FP navigation (*Figure 4K* and *Videos 9–12*).

## SlitC and SlitN are expressed as segregated clusters in the FP navigation path

Next, we studied the FP distribution patterns of PlxnA1 ligands. For Sema3B, we took advantage of our knock-in model of Sema3B-GFP fusion (*Arbeille et al., 2015*). Transverse sections were stained with anti-GFP antibody to amplify the

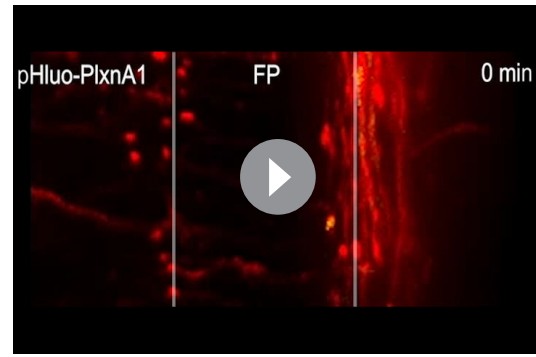

**Video 1.** pHluo-PlxnA1$_{WT}$ (*Videos 1* and *2*) and pHluo-PlxnA1$_{Y1815F}$ (*Videos 3* and *4*) are addressed to the cell surface of commissural growth cones during the FP navigation. White arrows point the growth cones during FP navigation. FP: floor plate.
https://elifesciences.org/articles/63205#video1

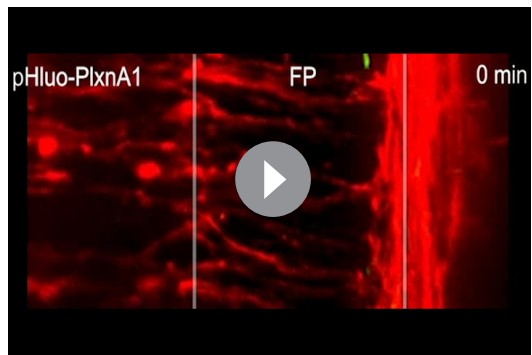

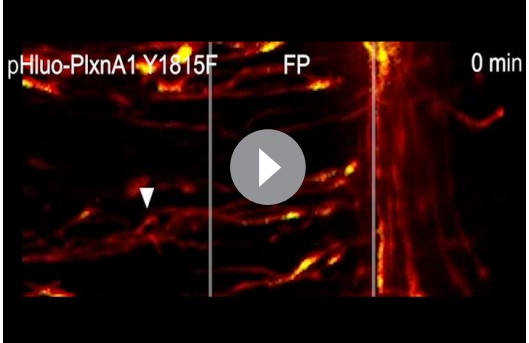

**Video 2.** pHluo-PlxnA1$_{WT}$ (*Videos 1* and *2*) and pHluo-PlxnA1$_{Y1815F}$ (*Videos 3* and *4*) are addressed to the cell surface of commissural growth cones during the FP navigation. White arrows point the growth cones during FP navigation. FP: floor plate.
https://elifesciences.org/articles/63205#video2

**Video 3.** pHluo-PlxnA1$_{WT}$ (*Videos 1* and *2*) and pHluo-PlxnA1$_{Y1815F}$ (*Videos 3* and *4*) are addressed to the cell surface of commissural growth cones during the FP navigation. White arrows point the growth cones during FP navigation. FP: floor plate.
https://elifesciences.org/articles/63205#video3

GFP signal, allowing its detection in the basal domain (*Figure 5A*). Quantitative analysis of the GFP distribution revealed that protein clusters distributed evenly in the FP, arranged in columns reflecting their localization along the basal processes.

Study of Slit fragments distribution is limited by the lack of antibodies allowing their individual detection and distinction from the full-length form. Thus, as in *Xiao et al., 2011*, we developed an alternative strategy based on fluorescent reporters, whose distribution should approximate that of endogenous fragments, since their physico-chemical properties might be highly similar. We constructed a plasmid encoding full-length Slit2 (Slit2-FL), fused to two distinct fluorescent proteins: Cerulean at its N-terminal part and Venus at its C-terminal part. Slit2-FL is visualized with both fluorophores overlapping (white signal). Upon cleavage, Slit2N is reported by Cerulean fluorescence (here in purple) and Slit2C by Venus (here in green) (*Figure 5B*). This construct was electroporated in the FP of E2 chick embryos (*Figure 5C*). Two days later, thick transverse sections were prepared and the FP observed by confocal microscopy. The FP domain was subdivided into three compartments along the dorso-ventral axis: (i) the 'apical' one, delimited by the central canal and the bottom of glial cells nuclei, (ii) the 'basal a' encompassing the dorsal half of the axons path, and (iii) the 'basal b', containing the ventral half of the axons path until the basal lamina (*Figure 5D*). We observed that glial cells display a massive white staining in their apical part, reflecting the presence of the uncleaved Slit2-FL and/or overlapping Slit2N and Slit2C fragments. Conversely, the basal compartments, in particular the most ventral one in which commissural axons preferentially path, showed higher degree of separation of the Slit2 fragments, as quantified by a Pearson coefficient evaluation (*Figure 5E*). Analysis of Cerulean (Slit2N) and Venus (Slit2C) fluorescence showed that both Slit2 fragments had a graded pattern, with higher levels in the basal a versus basal b compartment. Yet, Slit2C was significantly enriched in the basal b compartment, when compared to Slit2N (*Figure 5F*). Thus, although Slit2C and Slit2N patterns within the FP are close, they present some specificities that could result from distinct structural properties and/or from mechanisms regulating Slit2 prior to processing. To address this question, we examined whether Slit2C and Slit2N distribute similarly

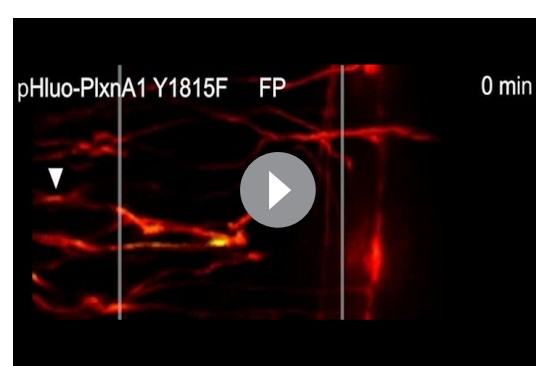

**Video 4.** pHluo-PlxnA1$_{WT}$ (*Videos 1* and *2*) and pHluo-PlxnA1$_{Y1815F}$ (*Videos 3* and *4*) are addressed to the cell surface of commissural growth cones during the FP navigation. White arrows point the growth cones during FP navigation. FP: floor plate.
https://elifesciences.org/articles/63205#video4

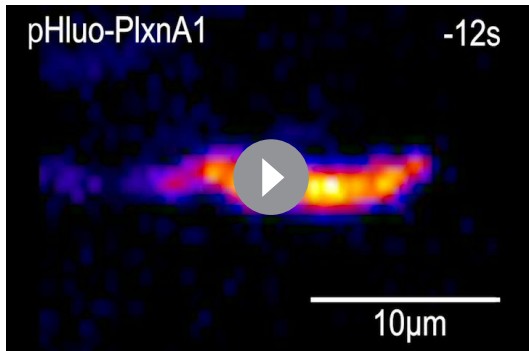

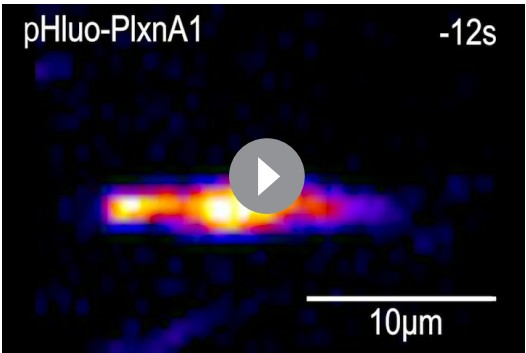

**Video 5.** Fluorescence recovery after photobleaching (FRAP) sequences of commissural growth cones in spinal cord open-books. The pHluo-receptor fluorescence in an area of 15–20 µm² covering the entire growth cone surface was bleached at 80–90%. The recovery was measured over a period of 17 min.
https://elifesciences.org/articles/63205#video5

**Video 6.** Fluorescence recovery after photobleaching (FRAP) sequences of commissural growth cones in spinal cord open-books. The pHluo-receptor fluorescence in an area of 15–20 µm² covering the entire growth cone surface was bleached at 80–90%. The recovery was measured over a period of 17 min.
https://elifesciences.org/articles/63205#video6

when issued from endogenous processing and when expressed individually. We constructed two plasmids encoding one fragment each in secreted fusion with a GFP (*Figure 5G*) and analyzed their pattern as performed previously. Interestingly, whereas Slit2N displayed similar distributions, Slit2C diffused massively in the axon path when issued from Slit2 cleavage (*Figure 5H–J*). Furthermore, Slit2C-GFP isolated fragment was more prone to deposition along the FP basal membrane than Slit2C processing product (*Figure 5K*). Thus, Slit2C and Slit2N distributions likely result from mechanisms regionalizing full-length Slit2 protein and its processing within the FP glial cell compartments.

## Proteolytic processing by PC2 convertase is required for proper patterns of Slit2N and Slit2C

Next, we investigated the potential contribution of Slit2-FL processing to the regionalization of Slit2 fragments in the native context. We constructed a plasmid encoding a dual-tagged Slit2-FL, fused to Cerulean in its N-terminal part and to Venus in its C-terminal part, having a deletion of the Slit cleavage site (amino acids 110–118-SPPMVLPRT-) (*Nguyen Ba-Charvet et al., 2001*). We verified by western blot that the expression of this Slit2-FL D in neuronal N2a cells was not compromised. As expected, the Cerulean-Slit2N and Slit2C-Venus fragments were detected in the Slit2-FL condition but not in the Slit2-FL D condition (*Figure 6A*). Dual-tagged Slit2-FL and Slit2-FL D were then electroporated in chicken embryos and transversal sections were analyzed by confocal microscopy. Notably, preventing Slit2 cleavage resulted in strong depletion of Slit2 fluorescent signal in the basal compartment in which commissural axons navigate (*Figure 6B–D*). This suggested that Slit2-FL is not addressed to the basal processes, rather distributed in the apical compartment of FP glia cells, where it is locally processed, generating fragments that are subsequently deposited on the basal processes. This confirmed that the cleavage is determinant for proper localization

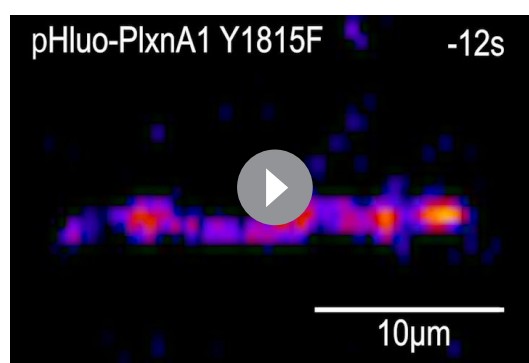

**Video 7.** Fluorescence recovery after photobleaching (FRAP) sequences of commissural growth cones in spinal cord open-books. The pHluo-receptor fluorescence in an area of 15–20 µm² covering the entire growth cone surface was bleached at 80–90%. The recovery was measured over a period of 17 min.
https://elifesciences.org/articles/63205#video7

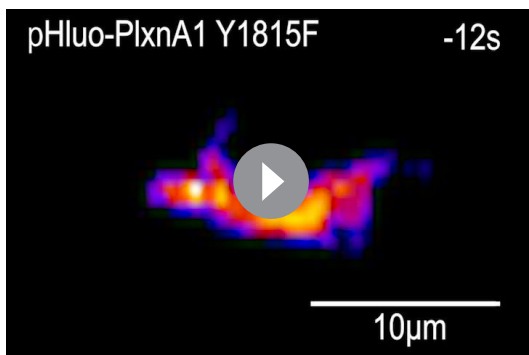

**Video 8.** Fluorescence recovery after photobleaching (FRAP) sequences of commissural growth cones in spinal cord open-books. The pHluo-receptor fluorescence in an area of 15–20 µm² covering the entire growth cone surface was bleached at 80–90%. The recovery was measured over a period of 17 min.
https://elifesciences.org/articles/63205#video8

of Slit2 fragments to the axon navigation path. While the protease responsible for Slit2 cleavage is still unknown in vertebrates, the pheromone convertase Amontillado has been shown to cleave Slit during muscles and tendons development in *Drosophila* (*Ordan and Volk, 2016*). Amontillado homolog in vertebrates is PC2, a proprotein convertase involved in the activation of endocrine peptides (*Smeekens et al., 1991*). We treated N2a cells transfected with dual-tagged Slit2-FL with chloromethylketone (CMK), a PC2 inhibitor, and found that it resulted in a loss of the Slit2C-Venus fragment and an accumulation of the FL form, as observed with Slit2-FL Δ. (*Figure 6E–G*). Thus, PC2 might either directly or indirectly be involved in Slit2 cleavage.

Then, we examined PC2 expression in transverse sections of E4 chicken embryos immunolabeled with an anti-PC2 antibody. We observed that PC2 was mostly expressed in the FP (*Figure 6H*). Signal quantification revealed that PC2 is enriched in the apical compartment with modest expression in the basal compartments of the FP (*Figure 6I and J*). Next, we investigated whether CMK injection to chicken embryos electroporated with dual-tagged Slit2-FL could alter Slit2 fragments distribution. Two injections of CMK (5 mM) and control vehicle (DMSO) were done, first following the electroporation and second 24 hr later. By quantifying the distribution of Slit2N and Sit2C, we found in both cases that the basal/apical signal was significantly reduced in the CMK condition compared with the control, indicating decrease of fragments deposition in the axon navigation path (*Figure 6K*). Thus overall, PC2 is expressed at an appropriate timing and location to mediate the cleavage of Slits and participate in the setting of their distribution patterns in the FP navigation path (*Figure 6L*).

## Y1815F mutation releases SlitC-mediated constrains imposed by the basal processes, resulting in abnormally plastic and exploratory growth cones during FP navigation

To get further insights into PlxnA1-SlitC mode of action, we compared PlxnA1$_{WT+}$ and PlxnA1$_{Y1815F+}$ growth cones morphologies in the FP. In chick open-books electroporated with pHluo-PlxnA1$_{WT}$ and pHluo-PlxnA1$_{Y1815F}$, we quantified growth cones of two classes of shape: oblong, modestly enlarged compared to the adjacent axon segment, and more complex, with visible protrusions and enlarged more than two times compared to the adjacent axon segment (*Figure 7—figure supplement 1*). We observed much more complex pHluo-PlxnA1$_{Y1815F+}$ growth cones (17%) than pHluo-PlxnA1$_{WT+}$ ones (5%) (*Figure 7—figure supplement 1*). Strikingly, we also observed these different morphologies in growth cones imaged using STED microscopy (*Figure 7—figure supplement 1*).

Next, we examined the behaviors of commissural neurons isolated from PlxnA1$_{Y1815F/Y1815F}$ embryos when exposed to SlitC and Sema3B. Our previous work showed that at basal condition mimicking the pre-crossing context, cultured commissural growth cones express low levels of PlxnA1 at their surface and are insensitive to SlitC and Sema3B. Treatment with GDNF, a cue expressed by FP cells, to mimic the FP crossing context, results in increased cell surface PlxnA1 and gain of sensitivity to subsequent SlitC and Sema3B bath application (*Delloye-Bourgeois et al., 2015*; *Charoy et al., 2012*). As expected from our previous overexpression experiments, the capacity of SlitC to collapse PlxnA1$_{Y1815F/Y1815F}$ commissural growth cones was significantly reduced, compared to that of Sema3B. Interestingly, in the SlitC/GDNF condition, we noted growth cones with atypical shapes: long filopodia, or contracted with numerous reminiscent filopodia, or visibly spread (*Figure 7—figure supplement 1*). This suggests that SlitC/PlxnA1 signaling might negatively regulate the complexity of growth cone morphologies.

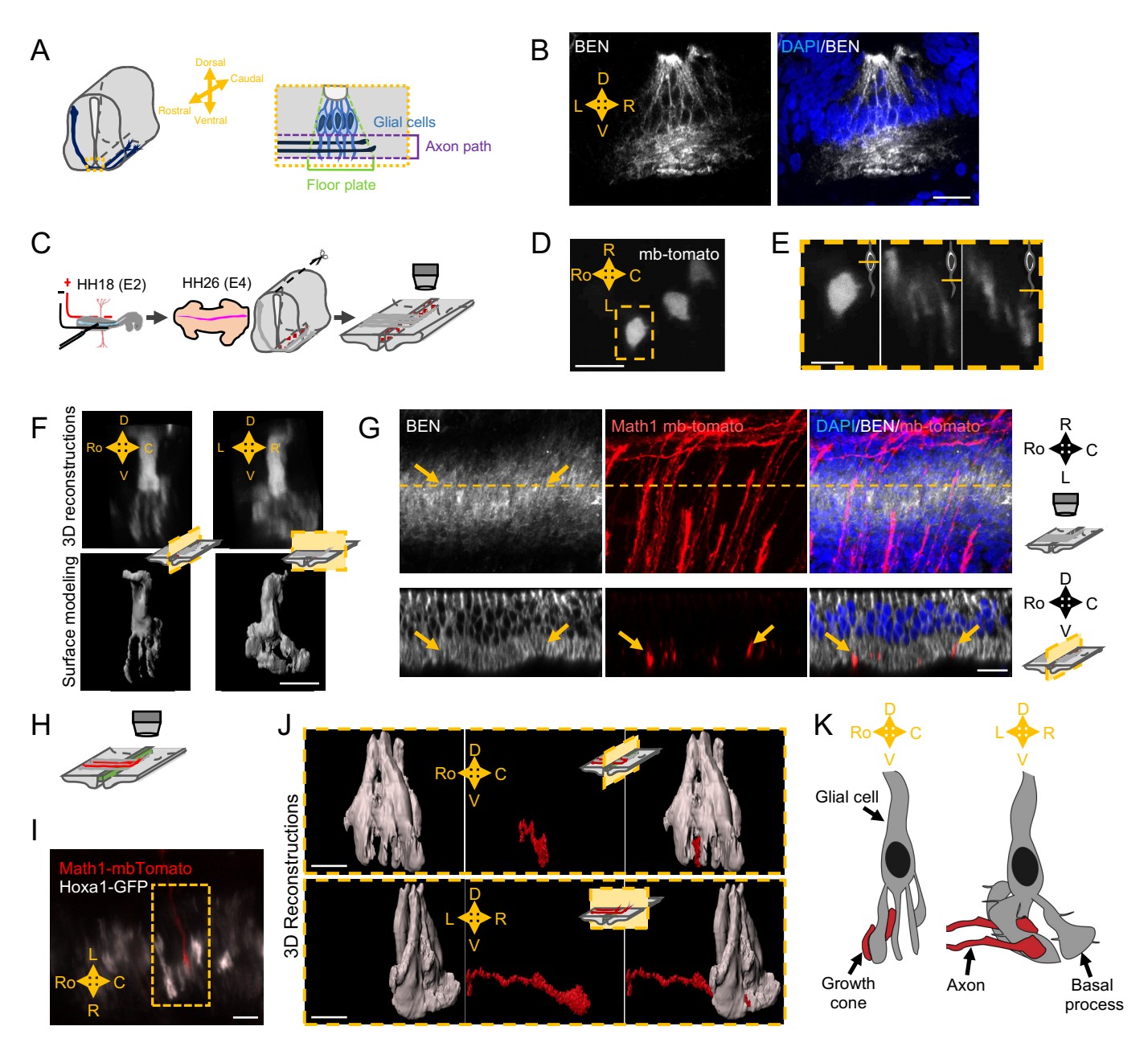

**Figure 4.** Spatial organization of the commissural axon navigation path. (A) Schematic drawings of a spinal cord at E12.5 when commissural axons cross the floor plate (FP) (left) and close-up of the FP (right) with glial cells (light blue) and crossing axons (dark blue). (B) Immunostaining of an E4 FP transverse section with FP specific BEN antibody (left) and merged with DAPI staining (right). (C) In ovo FP electroporation procedure. Sparse electroporated mbTomato electroporated cells are visualized in red. (D) Open-book imaging of E4 chick FP with sparse mbTomato electroporation. The dashed rectangle highlights a glial cell. (E) Close-up of the single glial cell observed in (D), at three different positions along the dorso-ventral axis, as shown by the schematic representation on the upper right corner of each image. (F) 3D reconstruction (upper) and surface modeling (lower) of a single FP cell seen in a sagittal (left) or transverse (right) section. (G) 3D reconstruction of axons (electroporated with mbTomato, red) navigating through the FP stained with DAPI (blue) and BEN (white). The yellow dashed line corresponds to the cut plane resulting in the sagittal optic section shown in the lower panel. Yellow arrows point growth cones intercalated between BEN labeled glial cell processes when they navigate the FP (see also *Videos 9–13*). (H) Schematic drawings of an open-book with a sparse electroporation of commissural axons and a broad FP electroporation. Two plasmids are used: Math1-mbTomato in commissural neurons and HoxA1-GFP in the FP. (I) Ventral longitudinal view from an open-book electroporated as described in (H). The yellow dashed rectangle delineates a close-up of a crossing growth cone electroporated with Math1-mbTomato (red) navigating along the basal feet of glial cells electroporated with Hoxa1-GFP (white). (J) Surface reconstruction of a growth cone electroporated with Math1-mbTomato (red) navigating along the basal feet of glial cells electroporated with Hoxa1-GFP (white) as seen in a sagittal section (upper) or

*Figure 4 continued on next page*

*Figure 4 continued*

transversal section (lower). (**K**) Schematic drawing of a glial cell and two axons crossing through its basal end-feet, in a sagittal view (left) or transverse view (right). Scale bars: 5 µm in (**D–G, I, and J**), 20 µm in (**B**).

To address the link between complex growth cones morphologies and aberrant trajectories, we recorded high-frequency time-lapses (one image every 8 min) of chick open-books electroporated with pHluo-PlxnA1$_{WT}$ or pHluo-PlxnA1$_{Y1815F}$. Individual growth cones navigating the FP were traced over time and scored according to two types of behavior: straight (maintained forward trajectory) and exploratory (deviations from the straight axis) (*Figure 7A*). Strikingly, we observed much more exploratory behaviors in the PlxnA1$_{Y1815F+}$ population (59%) than in the pHluo-PlxnA1$_{WT+}$ one (15%) (*Figure 7B* and *Videos 13–16*). Exploratory growth cones were either split or turned (deviated from their straight axis, orienting in rostral or caudal directions). While turned growth cones were found equally represented in both conditions, split PlxnA1$_{Y1815F+}$ growth cones were much more frequent than PlxnA1$_{WT}$ ones (49% and 9% respectively) (*Figure 7C and D*). We did not observe collateral elongating from split growth cones, which suggests that splitting is transient and followed by single growth cone reformation. Moreover, exploratory behaviors were not appearing after midline crossing or after FP exit, but were present from the onset of the FP navigation. To refine our analysis, we measured the width of individual growth cones over time, as a read-out of morphological plasticity and exploratory behavior. We found that PlxnA1$_{Y1815F+}$ growth cones were significantly larger than PlxnA1$_{WT+}$ ones. In addition, their increased exploratory behavior was also reflected in the higher variations of width over time when compared with PlxnA1$_{WT+}$ growth cones (*Figure 7E and F*).

## Discussion

Our results support a model whereby growth cone trajectories are actively kept straight over the entire FP navigation through reiterated contacts with SlitC spots of basal processes. PlxnA1 mutation altering perception of SlitC alleviates these constrains, which results in abnormally dynamic growth cones that actively explore the FP space. Such aberrant behavior compromises efficient and straight navigation across the FP leading to premature rostro-caudal deviation and turning back (*Figure 7G*). Thus, rather than pushing the growth cones and preventing them from turning back, SlitC constrains axon growth through aligned FP glia corridors, imposing them continuous forward direction.

### Specific subcellular architecture of the FP glia conforms the presentation of FP ligands to the navigating commissural axons

Characterization of guidance cues spatial distribution is only beginning. Recent work revealed that spinal cord progenitors that synthesize Netrin1, a major attractant for spinal cord commissural axons, take advantage of their basal process to redistribute Netrin at a more lateral position close to the basal lamina, where it promotes commissural axon navigation toward the FP (*Varadarajan and Butler, 2017*). Hence, despite extensive studies of midline crossing, very little is known on the FP tridimensional organization and guidance cues patterns. Early electronic microscopy images revealed close contacts between FP glia surfaces and axons, and suggested complex shapes of basal processes (*Yaginuma et al., 1991*; *Okabe et al., 2004*). Observations of FP cells in a β-galactosidase transgenic mouse line also pointed out the

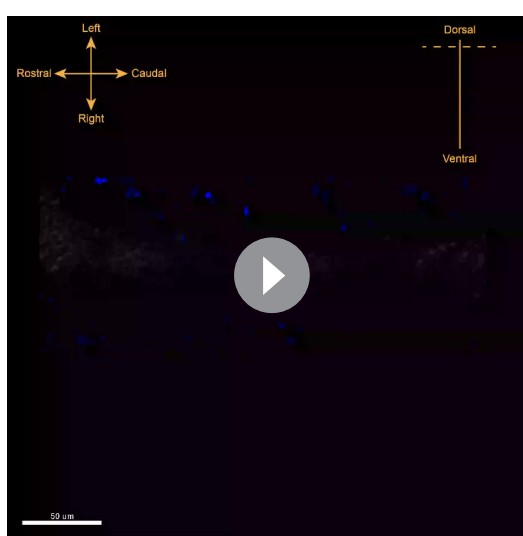

**Video 9.** 3D reconstruction with IMARIS software of axons navigating through the floor plate stained with DAPI (in blue) and BEN (in white) and with mbTomato electroporated in axons (in red).
https://elifesciences.org/articles/63205#video9

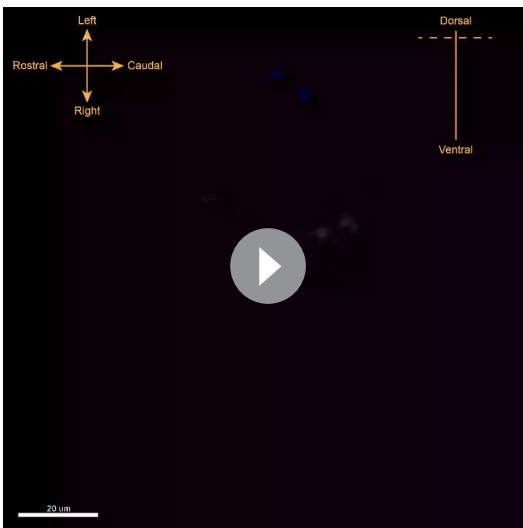

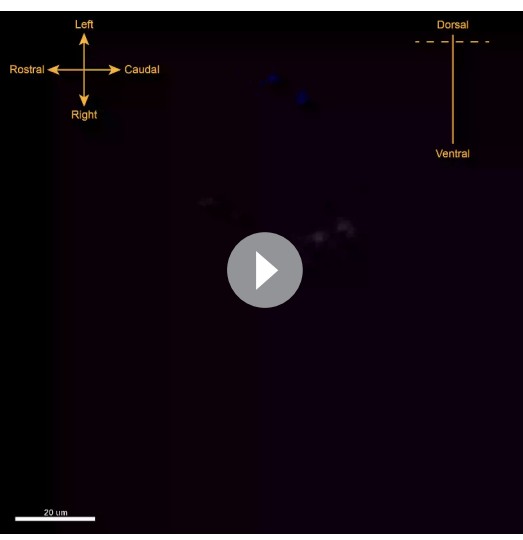

**Video 10.** Same reconstitution as in *Video 9* but on a limited slice of spinal cord cut in the rostro-caudal axis. https://elifesciences.org/articles/63205#video10

**Video 11.** The surface of a single axon from *Video 10* has been reconstructed. https://elifesciences.org/articles/63205#video11

complexity of basal processes (*Campbell and Peterson, 1993*). Our study extends these observations, revealing in more detail the 3D architecture of the commissural navigation path formed by the FP glia. A particularly striking observation was the asymmetric apico-basal polarity of the FP cell, with a single apical anchor and an enlarged basal process subdivided into several pillars. Notably, these pillars are oblong, flattened along the left–right axis. Moreover, in the dorso-ventral axis, their ramifications provide to the axons a net-shaped roof. Thus overall, the FP basal process ramifications form parallel narrow corridors shaping the axon navigation path. Interestingly these observations echo early and more recent studies reporting roles of glia specializations in axon navigation. In the developing brain, glia cells at the optic chiasm midline form a specific compartment, arranging a pal-

isade that crossing retinal axons traverse (*Marcus et al., 1995*). The path of callosal axons across the midline is also paved by radial glia processes (*Norris and Kalil, 1991*). More recently, a radial glia scaffold lying at the frontier between the striatum and the globus pallidum was discovered to guide corticospinal axons (*Kaur et al., 2020*).

Our study reveals that glia cells elaborate complex morphological specializations that distribute the guidance cues in the FP. Notably, we found that Slit2C, Slit2N, and Sema3B are prominently deposited in columnar patches covering the ramified basal processes of FP glia cells, with Slit2C and Slit2N forming distinct clusters. Unexpectedly, these observations reveal that the distribution pattern of Slit ligands is intimately controlled by the particular morphology of the FP glia cells. Our analysis also revealed that Slit2-FL protein concentrates in the soma and its cleavage is a prerequisite for Slits deposition in the commissural axon path. Thus, unlike in *Drosophila* midline guidance for which Slit processing was reported to be dispensable (*Coleman et al., 2010*), FL-Slits likely play

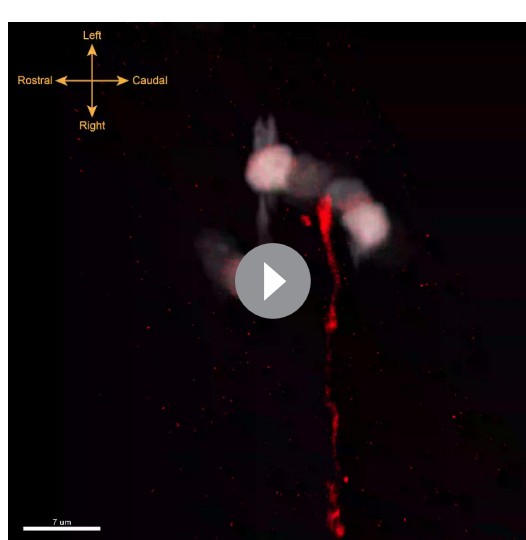

**Video 12.** 3D reconstruction from a Math1-mbTomato and Hoxa1-GFP electroporation. A single Math1-mbTomato (in red) electroporated axon is navigating through the basal end-feet of a Hoxa1-GFP (in white) electroporated floor plate cell. https://elifesciences.org/articles/63205#video12

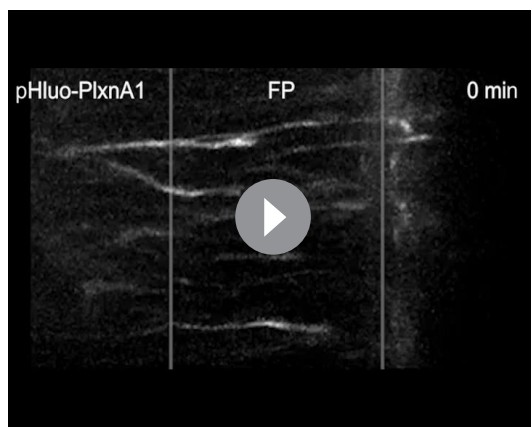

**Video 13.** Sequences of time-lapse movies of chick open-books at fast time intervals (8 min) illustrating the navigation behaviors of PlxnA1$_{WT}$ growth cones (*Videos 13* and *14*) and PlxnA1$_{Y1815F}$ growth cones (*Videos 15* and *16*).

https://elifesciences.org/articles/63205#video13

limited guidance role for commissural axons. This is also consistent with previous studies of the process of muscle anchorage to tendons in *Drosophila* that reported the importance of Slit processing for subsequent distribution of the fragments and proper functional outcome (*Ordan and Volk, 2016*). Our data suggest that the pro-protein convertase PC2, the vertebrate homolog of Amontillado, that we found able to cleave Slit2-FL and expressed in the FP, contributes to the generation of Slit2C and SlitN fragments distribution. Consistently, its in vivo pharmacological inhibition resulted in decreased fragment deposition in the axon navigation path. Whether in ovo administration of PC2 inhibitor resulted in alteration of commissural axon navigation remains to be demonstrated. Addressing this question would require temporally accurate abrogation of Slit cleavage to achieve Slit fragment depletion at the time the axons navigate the FP, which is technically difficult to achieve.

SlitN and SlitC fragments were reported to have different cell association properties (*Brose et al., 1999*). The molecular scaffold that localizes the ligands in the FP is still to be explored. Extracellular matrix components and their interactors are well acknowledged for their critical role in setting molecular patterns (*Walker et al., 2018*). Interestingly, in the *zebrafish* developing brain, apical end feet of radial glial cells extend at the surface of the tectum, organizing through collagen IV the lamination of the neuropil by exposing Slits and heparan sulfate proteoglycans to axon terminals (*Xiao et al., 2011*). Morphogenesis of the *Drosophila* heart tube lumen was found to rely on polarized localization of the collagen XV/XVIII orthologue Multi-Plexin forming a macro-complex with Slit (*Harpaz et al., 2013*). In the context of midline crossing, loss of the heparan sulfate carrier (heparan sulfate proteoglycan) Syndecan results in modification of Slit distribution and activity in *Drosophila* (*Johnson et al., 2004*). In the vertebrate spinal cord, dystroglycan was identified as a Slit binding partner, localizing the proteins in the basal lamina through non-cell autonomous action (*Wright et al., 2012*; *Lindenmaier et al., 2019*). Consistently, we detected the presence of SlitC, whose sequence contains dystroglycan binding motif, in the basement membrane, as well as that of SlitN, possibly sequestrated by additional components of the FP basal lamina. For example, proteolysis of the extracellular matrix protein F-spondin, reported to contribute to commissural axon guidance (*Burstyn-Cohen et al., 1999*), was found to generate two fragments, among which one is deposited at the basement membrane (*Zisman et al., 2007*).

## Commissural axon navigation is driven by restriction of growth cone exploration resulting from reiterated contacts with ligands deposited on the FP basal processes

The mode of action by which FP cues exert their effect is puzzling. Proper FP crossing is thought to rely on a balance of positive and negative forces that support growth cone attachment, motility, and direction (*Stoeckli et al., 1997*; *Fitzli et al., 2000*; *Gore et al., 2008*; *de Ramon Francàs et al., 2017*). Moreover, studies of midline crossing in *Drosophila* inspired the general view that commissural axons acquire sensitivity to local repellents after midline crossing and are pushed toward the contralateral side (*Evans and Bashaw, 2010*). Consistently, differential pre- and post-crossing sensitivities were reported in numerous experimental paradigms (*Zou et al., 2000*; *Delloye-Bourgeois et al., 2015*; *Nawabi et al., 2010*) but such pushing action of the cues has not been clearly established.

Our findings argue for a very different mode of action. With our analysis of PlxnA1$_{Y1815F}$ model and live imaging paradigms, we show that SlitC acts by restricting the exploratory capacity of commissural growth cones, which sort PlxnA1 when they enter the FP (*Pignata et al., 2019*). This function is served by the particular morphology of the FP glia cells that might impose repeated contacts

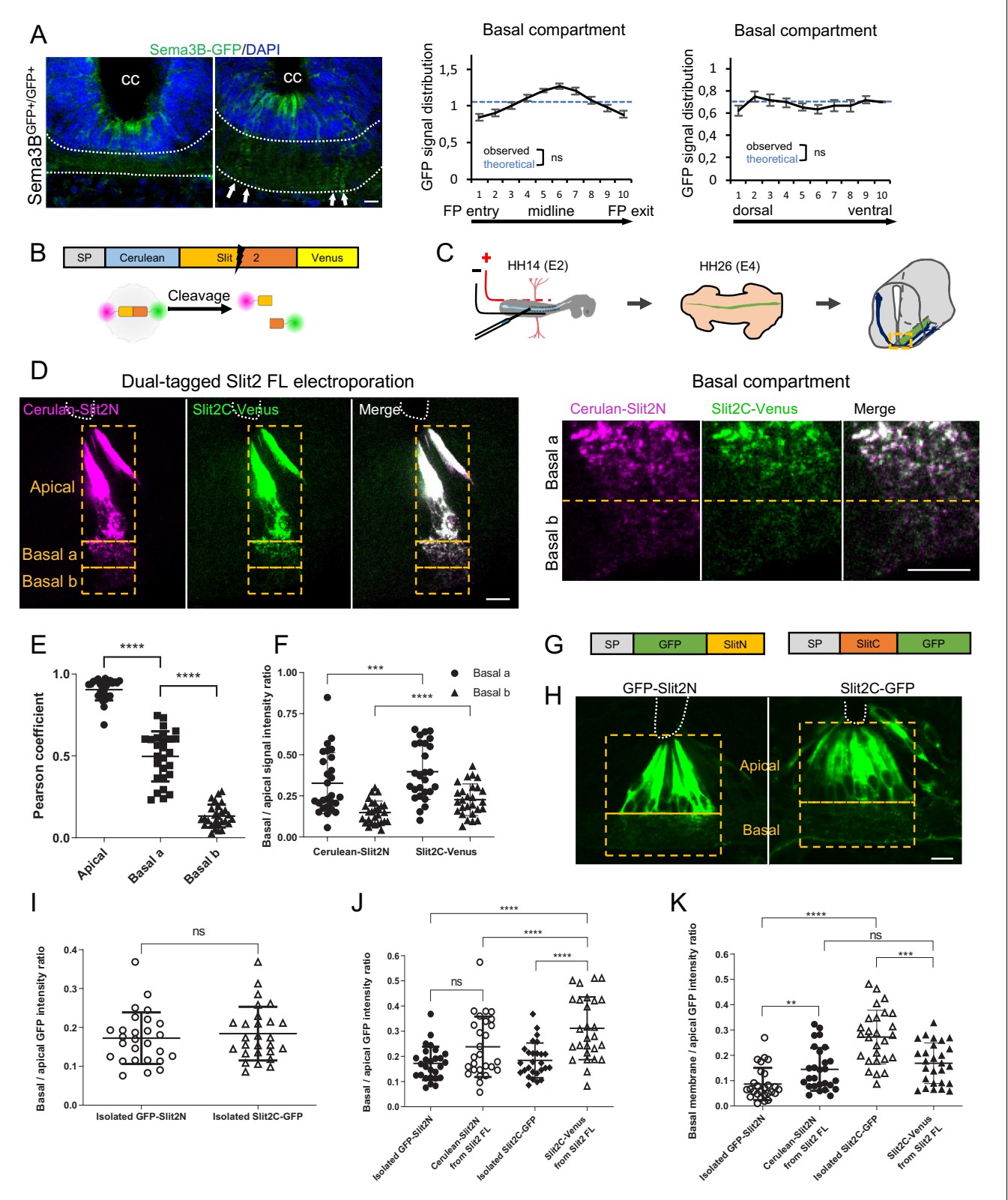

**Figure 5.** Slit2N and Slit2C decorate the floor plate (FP) glia basal processes and have distinct diffusion properties conditioned by Slit2-FL processing.
(**A**) Transverse section of Sema3B-GFP homozygous E12.5 embryo stained with DAPI. Endogenous green fluorescent protein (GFP) reports Sema3B expression and is quantified in the left–right axis of the FP (left) and in the dorso-ventral axis (right). The axon path is delineated by white dashed lines and the white arrows indicate the deposition of Sema3B-GFP along the glial cell processes. Data are shown as the mean ± s.e.m., KS test has been

*Figure 5 continued on next page*

*Figure 5 continued*

applied, n.s. (**B**) Schematic drawings of dual-tagged Slit2 construct and activity. (**C**) In ovo FP electroporation procedure. (**D**) 80 μm transverse section of E4 chick spinal cord FP electroporated with dual-tagged Slit2 (left) and close-up of the basal domain (right). (**E**) Pearson coefficients quantify the degree of colocalization of Cerulean Venus in FP electroporated with dual-tagged Slit2, through three compartments: apical, basal a, and basal b, as delimited by the dashed rectangle in (**D**) (N = 3 embryos, three sections per embryo, three images analyzed per section). (**F**) Intensity ratio of the basal compartment over the apical compartment for Cerulean and Venus in FP electroporated with dual-tagged Slit2. (**G**) Schematic drawings of Slit2 isolated fragments fused to GFP. (**H**) 80 μm transverse sections of E4 chick spinal cord FP electroporated with isolated Slit2 fragments fused to GFP. The apical and basal compartment are delineated with yellow dashed lines. (**I**) Intensity ratio of the basal compartment over the apical compartment for GFP in FP electroporated with either Slit2N-GFP or Slit2C-GFP (N = 3 embryos, three sections per embryo, three images analyzed per section). (**J**) Comparison between the basal/apical intensity ratio of Slit2 isolated fragments compared to the basal/apical intensity ratio of Slit2N and Slit2C fragments generated by the cleavage of dual-tagged Slit2-FL. (**K**) Comparison between the basal membrane/apical domain intensity ratio of Slit2 isolated fragments compared to the basal membrane/apical domain ratio of Slit2N and Slit2C fragments generated by the cleavage of dual-tagged Slit2-FL. Data are shown as the mean ± s.d. in (**E, F, and I–K**), Student's t-test has been applied, ns: non-significant, ***: $p<0.001$, ****: $p<0.0001$. Scale bars: 10 μm in (**A, D, and H**).

The online version of this article includes the following source data for figure 5:

**Source data 1.** Quantification of Sema3B expression in the left-right axis of the FP and in the dorso-ventral axis (A).
**Source data 2.** Pearson coefficients quantify the degree of colocalization of Cerulean Venus in FP electroporated with dual-tagged Slit2 (E).
**Source data 3.** Intensity ratio of the basal compartment over the apical compartment for Cerulean and Venus in FP electroporated with dual-tagged Slit2 (F).
**Source data 4.** Intensity ratio of the basal compartment over the apical compartment for GFP in FP electroporated with either Slit2N-GFP or Slit2C-GFP (I).
**Source data 5.** Comparison between the basal/apical intensity ratio of Slit2 isolated fragments compared to the basal/apical intensity ratio of Slit2N and Slit2C fragments generated by the cleavage of dual-tagged Slit2-FL (J).
**Source data 6.** Comparison between the basal membrane/apical domain intensity ratio of Slit2 isolated fragments compared to the basal membrane/apical domain ratio of Slit2N and Slit2C fragments generated by the cleavage of dual-tagged Slit2-FL (K).

of the growth cones with spots of ligands spanning the navigation path. Our functional analyses are all consistent with SlitC mode of action restricting the formation or the stabilization of peripheral growth cone protrusions to constrain them in a non-exploratory state, thus favoring forward growth direction. Such effect is also reflected in the shape of WT crossing growth cones, which we observed are rostro-caudally thin and dorso-ventrally elongated. Notably, PlxnA1$_{Y1815F+}$ growth cones have a much more complex morphology than WT growth cones, and they also sense the environmental content much more actively. In drastic cases, some were even seen to turn back on themselves, although they were more frequently directed toward the contralateral side. In recent work, axon growth was modeled using microcontact printing culture devices developed to examine individual growth cone responses to dots of guidance molecules (*Ryu et al., 2019*). Axons were challenged to grow on micropatterned surfaces consisting in uniform Sema3F substrate interrupted by permissive dots. They were found able to efficiently extend in such surfaces, having, in addition, a straight trajectory resulting from jumping from one permissive dot to the next one. Interestingly, perturbing the distance between dots or their size modified the shape of the axons by disorganizing their cytoskeleton and their trajectory, inducing curved growth patterns (*Ryu et al., 2019*). Thus, a 'salt and pepper' context as in the FP with repulsive ligands localized in spots and intercalated with permissive regions might build an appropriate topographic environment for limiting the possibilities of growth deviations.

Such constrain of growth cone exploration might also be required for counteracting other local coincident guidance forces that would disturb axon trajectories during FP crossing. In our recent work, we found in live imaging that the exploration is increased when growth cones start navigating the second FP half, although we observed they still keep a clear straight growth (*Pignata et al., 2019*). This exploratory behavior could reflect an increasing sensitivity to FP-derived Shh and Wnt rostro-caudal gradients that were found to drive the longitudinal turning after the crossing (*Zuñiga and Stoeckli, 2017*). The exact timing at which the axons become sensitive to these gradients is unclear. In Robo3 knockout embryos, commissural axons totally fail to cross the FP but turn longitudinally in the ipsilateral side. This argues that non-crossing commissural axons can get sensitized to longitudinal gradients, which also reflects the need for guidance constrains to counteract premature turning during the FP navigation (*Friocourt and Chédotal, 2017*). One function of PlxnA1/SlitC signaling would thus be to prevent premature longitudinal turning until FP exit. This is

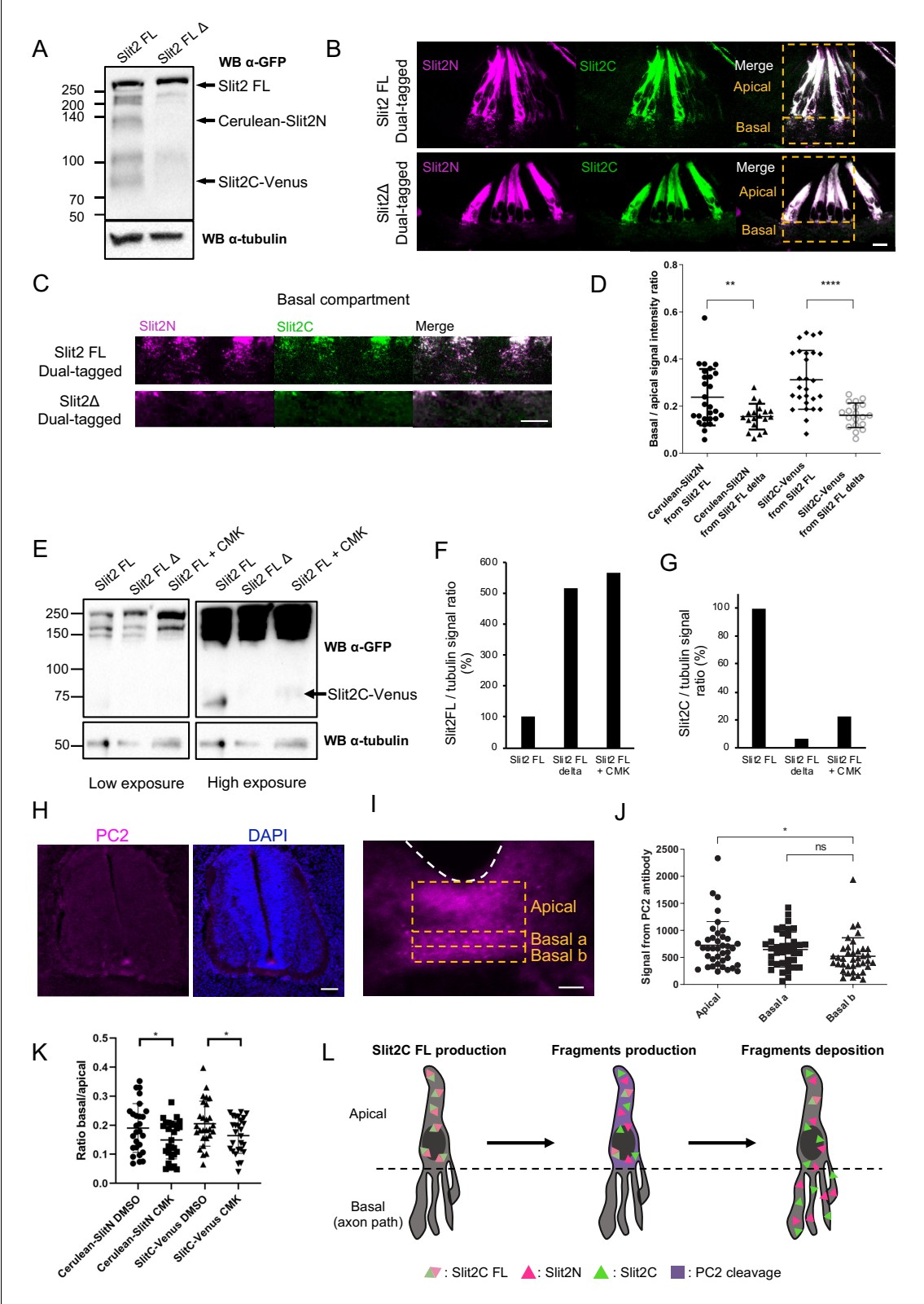

**Figure 6.** Slit2-FL cleavage plays a role in the proper diffusion of the protein and is dependent on PC2. (**A**) Western blot detection of Cerulean and Venus in N2a cells transfected with either dual-tagged Slit2-FL or the uncleavable dual-tagged Slit2-FL Δ. An anti-green fluorescent protein (GFP) antibody was used, which recognizes Cerulean and Venus, two GFP derived fluorescent proteins. Protein sizes are in kDa. (**B**) Thick transverse sections of E4 chick spinal cord floor plate (FP) electroporated with dual-tagged Slit2-FL or dual-tagged Slit2-FL Δ (uncleavable form deprived from the cleavage

*Figure 6 continued on next page*

*Figure 6 continued*

site generating Slit2N and Slit2C fragments). (**C**) Close-up of the basal domains from (**B**). (**D**) Intensity ratio of the basal compartment over the apical compartment for Cerulean and Venus in FP electroporated with dual-tagged Slit2-FL or dual-tagged Slit2-FL Δ. (**E**) Western blot detection of Cerulean and Venus in N2a cells transfected with either dual-tagged Slit2-FL or the uncleavable dual-tagged Slit2-FL Δ and treated with PC2 inhibitor CMK. Tubulin is used as a loading control. (**F and G**) Ratio of the intensity of the Slit2-FL band (**F**) or the Slit2C band (**G**) over the intensity of tubulin band. (**H**) Immunofluorescent labeling of E4 chick spinal cord transverse sections using an antibody targeting PC2 (left panel) and DAPI (right panel). (**I**) Close-up of the PC2 labeling in the FP. (**J**) Quantification of the PC2 labeling in the three compartments delineated by yellow dashed lines. (**K**) Quantification of Slit2C and Slit2N basal/apical signal in control DMSO and CMK injected embryos after dual-tagged Slit2-FL electroporation. (**L**) Schematic representation of Slit2 fragments generation and deposition from the cleavage of Slit2-FL by PC2 in FP glial cells. Data are shown as the mean ± s.d. in (**D and J**) and Student's t-test has been applied. ns: non-significant, *: p<0.05, ***: p<0.001. Scale bars = 10 µm in (**B, C, and I**), 80 µm in (**H**).

The online version of this article includes the following source data for figure 6:

**Source data 1.** Intensity ratio of the basal compartment over the apical compartment for Cerulean and Venus in FP electroporated with dual-tagged Slit2-FL or dual-tagged Slit2-FL Δ (D).

**Source data 2.** Ratio of the intensity of the Slit2-FL band over the intensity of tubulin band (F).

**Source data 3.** Ratio of the intensity of the Slit2C band over the intensity of tubulin band (G).

**Source data 4.** Quantification of the PC2 labeling in the three compartments delineated by yellow dashed lines (J).

**Source data 5.** Quantification of Slit2C and Slit2N basal/apical signal in control DMSO and CMK injected embryos after dual-tagged Slit2-FL electroporation (K).

fully consistent with our observation both in chick and mouse open-books that a significantly higher proportion of axons undergo early rostro-caudal turn during the FP navigation, when deficient for SlitC responsiveness.

Our model supports that despite their early sensitivity to Slit fragments, commissural axons are still able to enter the FP. This might result from the distribution of the cues concentrating them in discrete spots deposited on the glial surfaces forming the corridors, rather than in diffuse distribution in the environment. Moreover, signaling locally promoting the entry of the axons could be concomitantly provided by FP cells, for example involving cell adhesion molecules, as reported in previous work (*Stoeckli et al., 1997*).

## Tyrosine 1815 mutation alters the membrane mobility of PlxnA1 receptor

The present study shows that mutating the Y1815 residue strongly impacts on the distribution of PlxnA1 at the cell membrane. First, we observed in mouse commissural axons navigating the FP in PlxnA1$_{-/-}$ open-books that the PlxnA1$_{Y1815F}$ cell surface pool expanded to the adjacent axon shaft, whereas it was much more restricted to the growth cone compartment in the WT context. Second, when expressed in chicken open-books, using STED imaging, PlxnA1$_{Y1815F}$ was observed to distribute at the membrane of the entire growth cone, not accumulating at the front as normally observed for PlxnA1$_{WT}$ (*Pignata et al., 2019*). Third, FRAP experiments revealed an increased membrane mobility of the PlxnA1$_{Y1815F}$ receptor pool, compared with the WT one, with a pattern of fluorescence recovery suggesting that exocytosis-mediated sorting is in contrast unaffected by the mutation. Overall, this raises the interesting possibility that PlxnA1 baring Y1815F mutation lacks or abnormally interacts with a protein whose function in stabilizing the receptor at the growth cone membrane is indispensable for SlitC signaling. Structural analysis established the ability of PlxnA1 to engage its extracellular domain in dimeric complexes (*Janssen et al., 2010*; *Kong et al., 2016*). Thus, Y1815 could be required for allowing the formation of macro-complexes of PlxnAs at the growth cone membrane. Alternatively, interactions of PlxnA1 cytoplasmic residues with signaling partners could result in abnormal PlxnA1 receptor dynamics. Our previous and present analyses indicate that PlxnA1$_{Y1815F/+}$ embryos have a recrossing phenotype while PlxnA1$^{+/-}$ ones do not (*Delloye-Bourgeois et al., 2015*). This could reflect that in heterozygous context the mutated receptor alters the activity of the WT one, thus aggravating the deficiency. Although we have no evidence from our competition assay supporting this model, further investigations are needed to clarify this issue, for example by addressing whether PlxnA1$_{Y1815F}$ and PlxnA1$_{WT}$ receptors could form heterodimers that would trap the functional receptor and limit its availability for SlitC signaling.

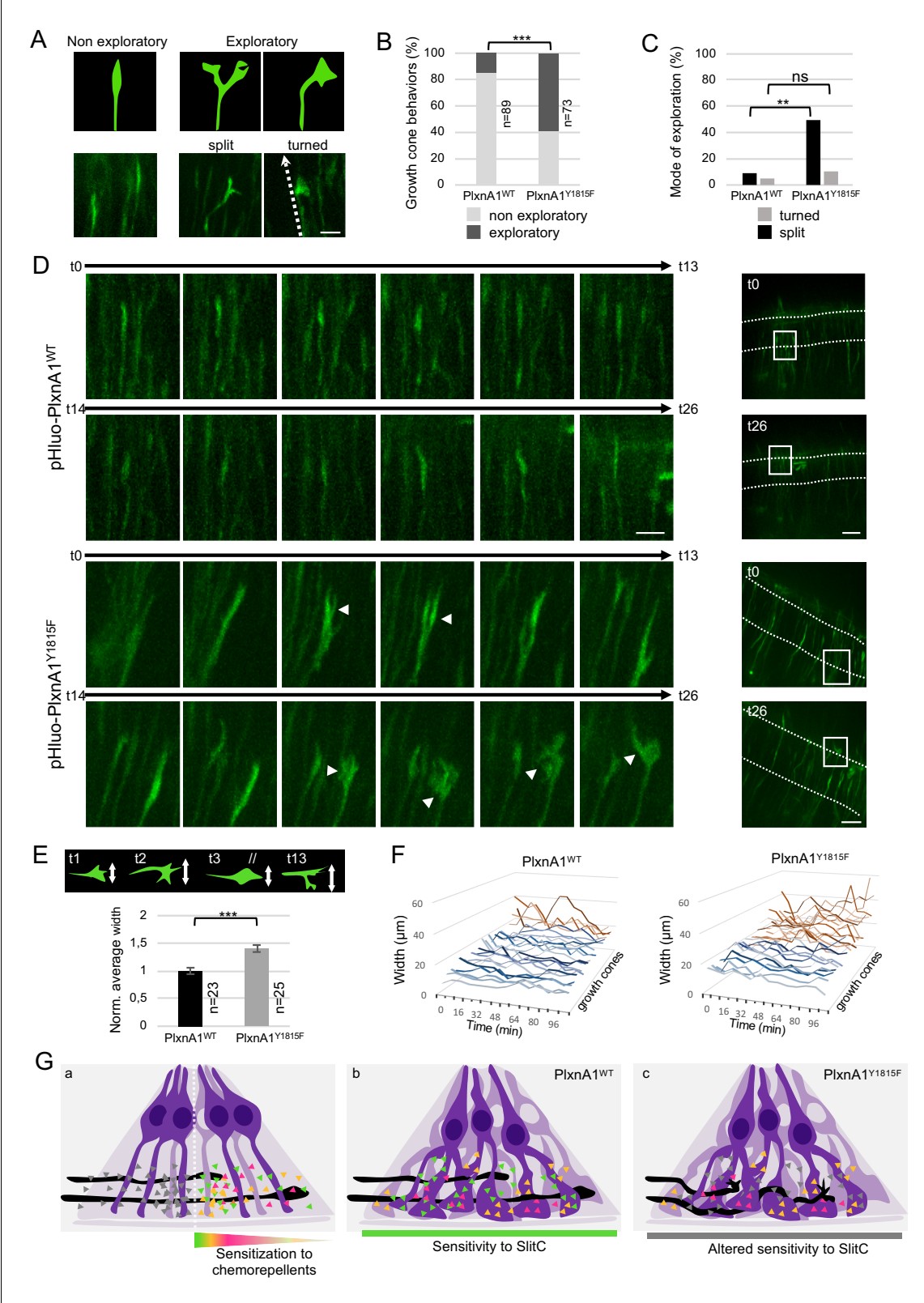

**Figure 7.** PlxnA1$_{Y1815F}$ commissural growth cones have an increased exploratory behavior exerted through morphological split. (**A**) Analysis of growth cone behaviors with fast time-lapse sequences. Schematic representation of growth cone categories and microphotographs of representative growth cones. (**B**) Histogram of the quantification of exploratory growth cones in PlxnA1$^{WT}$ and PlxnA1$_{Y1815F}$ open-books (pHluo-PlxnA1$_{WT}$, N = 5 electroporated embryos, 89 growth cones; pHluo-PlxnA1$_{Y1815F}$, N = 3 electroporated embryos, 73 growth cones). Chi-squared test has been applied,

*Figure 7 continued on next page*

*Figure 7 continued*

\*\*\*: p<0.001. (C) Histogram depicting the mode of exploration adopted by PlxnA1$_{WT}$ and PlxnA1$_{Y1815F}$ growth cones. The percentage was calculated over the total growth cone population (pHluo-PlxnA1$_{WT}$, N = 5 electroporated embryos, 89 growth cones; pHluo-PlxnA1$_{Y1815F}$, N = 3 electroporated embryos, 73 growth cones). Chi-squared test has been applied between non-exploratory and split populations (\*\*: p<0.01), and between non-exploratory and turned (ns: non-significant). (D) Time-lapse sequences of individual growth cones navigating the floor plate (FP). The right panels show growth cone positions at time 0 and time 26. Time interval: 8 min. (E) Quantification of growth cone width reporting their exploratory behavior during the FP navigation in PlxnA1$_{WT}$ and PlxnA1$_{Y1815F}$ open-books (pHluo-PlxnA1$_{WT}$, N = 4 electroporated embryos, 23 growth cones; pHluo-PlxnA1$_{Y1815F}$, N = 3 electroporated embryos, 25 growth cones). Data are shown as the mean ± s.e.m., Student's t-test has been applied, \*\*\*: p<0.001. (F) Histograms of individual growth cone width (µm) from t = 0 min to t = 122 min (pHluo-PlxnA1$_{WT}$, N = 4 electroporated embryos, 23 growth cones; pHluo-PlxnA1$_{Y1815F}$, N = 3 electroporated embryos, 25 growth cones). (G) Current and proposed novel model of the mechanisms ensuring proper midline crossing of spinal cord commissural axons. Scale bars: 10 µm in (A and D, left), 50 µm in (D, right).

The online version of this article includes the following source data and figure supplement(s) for figure 7:

**Source data 1.** Quantification of exploratory growth cones in PlxnA1$^{WT}$ and PlxnA1$_{Y1815F}$ open-books (B and C).

**Source data 2.** Quantification of individual growth cone width (F).

**Figure supplement 1.** PlxnA1$_{Y1815F}$ mutation confers to commissural growth cones navigating the floor plate (FP) a complex morphology.

**Figure supplement 1—source data 1.** Comparative analysis of the proportion of complex growth cones observed after PlxnA1$_{WT}$ and PlxnA1$_{Y1815F}$ electroporation (B).

**Figure supplement 1—source data 2.** Histogram depicting the proportion of collapsed growth cones in the different experimental conditions (E).

## Is PlxnA1-SlitC signaling the only player of the guidance program ensuring straight trajectory across the FP?

Interestingly, *astray* mutation in the zebrafish affecting the Robo2 gene was found to result in altered behaviors of navigating retinal axons, resembling those we report here for spinal commissural axons defective for PlxnA1-SlitC signaling. In astray mutants, retinal growth cones coursing toward the midline and away after crossing are larger and less streamlined than WT ones, also abnormally deviating their trajectory and not correcting their errors as WT ones do. Analysis of Slit patterns indicated that these ligands flank the path, suggesting they act by channeling the growth of retinal axons in the ventral telencephalon (*Hutson and Chien, 2002*). In the mouse spinal cord, recrossing phenotypes have been observed when PlxnA1 and Slit1-3, but not Robo1/2 or Sema3B, are inactivated and we show here that it is manifested when PlxnA1 fails to properly mediate SlitC activity. This does not exclude that the Slit/Robo signaling also contributes to keep forward axon navigation. The lack of recrossing in context of Robo1/2 loss could be due to the specific temporal and spatial pattern of Robo1 receptor at the growth cone surface. According to our previous work (*Pignata et al., 2019*), growth cones navigating the first FP half express PlxnA1 but not yet Robo1, while in the second FP half, they express both. Robo2 was found to be sorted only during the post-crossing longitudinal navigation (*Pignata et al., 2019*). Thus, PlxnA1 and Robo loss cannot have the same outcome, whether or not both receptors have redundant functions. PlxnA1 deletion results in a situation where growth cones lack both Robo1 and PlxnA1 when they enter the FP and navigate the first FP half, whereas Robo deletion will be manifested later when growth cones navigate the second FP half and express PlxnA1. Moreover, recrossing is likely to only be the more drastic aberrant behavior. Interestingly in Robo1/2$^{-/-}$ embryos, cases of commissural axons aberrantly oriented within the FP were reported that could reflect alleviation of straight growth constrains (*Long et al., 2004*).

Overall, our study raises a guidance model of midline crossing that might represent a general

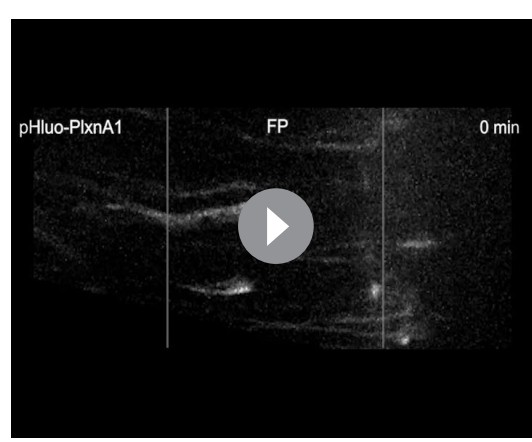

**Video 14.** Sequences of time-lapse movies of chick open-books at fast time intervals (8 min) illustrating the navigation behaviors of PlxnA1$_{WT}$ growth cones (*Videos 13* and *14*) and PlxnA1$_{Y1815F}$ growth cones (*Videos 15* and *16*).
https://elifesciences.org/articles/63205#video14

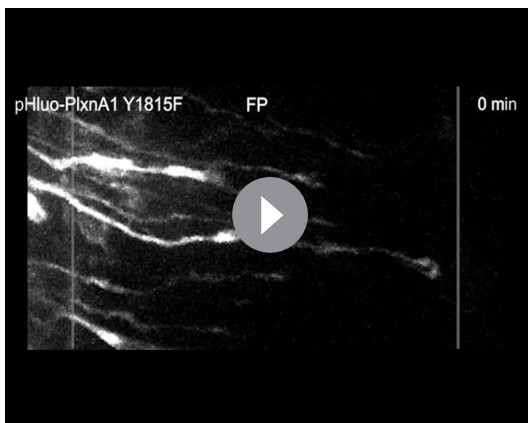

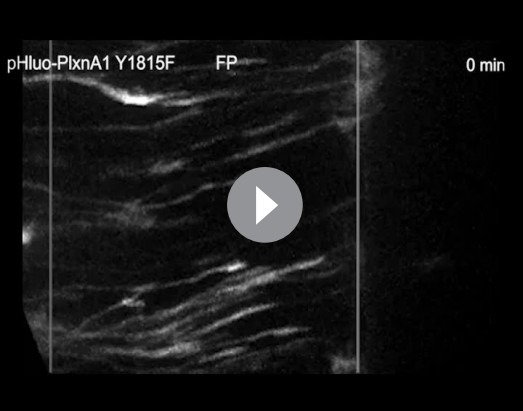

**Video 15.** Sequences of time-lapse movies of chick open-books at fast time intervals (8 min) illustrating the navigation behaviors of PlxnA1WT growth cones (*Videos 13* and *14*) and PlxnA1Y1815F growth cones (*Videos 15* and *16*).

https://elifesciences.org/articles/63205#video15

**Video 16.** Sequences of time-lapse movies of chick open-books at fast time intervals (8 min) illustrating the navigation behaviors of PlxnA1WT growth cones (*Videos 13* and *14*) and PlxnA1Y1815F growth cones (*Videos 15* and *16*).

https://elifesciences.org/articles/63205#video16

mechanism for the navigation at choice points. Continuously limiting growth cone exploration through short-range signaling that maintains their trajectory straight can efficiently counteract coincident guidance cues destined for the next step, preventing them to induce premature directional changes until intermediate target navigation is completed.

## Materials and methods

### Generation of the PlxnA1$_{Y1815F}$ mouse line and genotyping

The PlxnA1$_{Y1815F}$ mouse line was generated by the Mouse Clinical Institute (Strasbourg, France). Mice were back-crossed to obtain PlxnA1$_{Y1815F}$ in a C57/BL6 background. Mice were hosted either in SPF (ALECS SPF, Lyon, France) or conventional (SCAR, Lyon, France) animal facility with ad libitum feeding. This study was covered by a Genetically Modified Organisms approval (number 561, French ministry for Research) and a local ethical comity (CECCAPP, Lyon). Genotyping was performed on dissected tails lysed in NaOH 1M solution for 25 min at 95°C and digested overnight with Proteinase K at 56°C with the following primers: 5′-CTT ATA GAT CTA GAC AGG CAG GGA GAC CAT-3′ and 5′-CGG TTG TCT TCT CGA GTA TCA CAC TCC TA-3′. The PCR kit used was FastStrat PCR Master (Roche, 04710436001). The amplification product from mutated allele has a 341 bp size and the one obtained from a wild-type allele is 262 bp. Genotyping of PlxnA1$_{-/-}$ was done as in *Yoshida et al., 2006*.

### Molecular biology

FL mouse pHluo-PlxnA1 was generated by introducing in Nter the coding sequence of the pHluo cloned from a vector encoding GABA A pHluo-GFP (*Jacob et al., 2005*). FL mouse pHluo-PlxnA1$_{Y1815F}$ was obtained by directed mutagenesis using the In-fusion HD Cloning Plus Kit (638909, Ozyme) with the following primers: 5′-AGG TAC TTT GCT GAC ATT GCC and 5′-GTC AGC AAA GTA CCT TTC AAC. The pH dependency of the fluorescence was validated as in *Pignata et al., 2019*.

Dual-tagged Slit2 was designed using the human Slit2 sequence (NM_001289135.2) and ordered from Genscript Biotech. Briefly, Cerulean was inserted after Slit2 signal peptide (MRGVGWQMLSLG LVLAILNKVAPQACPA) and Slit2 FL sequence. Venus was fused to the C-terminal part of Slit2. Dual-tagged Slit2 Δ construct was obtained by Quickchange mutagenesis procedure (Agilent), deleting nine amino acids (SPPMVLPRT) at the cleavage site. The primer used for the mutagenesis were 5′-CTT GTT CTG TGA GTTT AGC CCC TGT GAT AAT TTT G-3′ and 5′-CAA AAT TAT CAC AGG GGC

T AAA CTC ACA GAA CAAG-3′. Both constructs were cloned into a PCAGEN vector using NotI and EcoRV digestion. The Hoxa1-GFP plasmid was a kind gift of Esther Stoeckli.

The fusion of Slit2N and Slit2C with GFP was performed in order to express the fusion proteins resulting from the cleavage of Slit2 Dual-tagged construct: GFP-SlitN (GFP in Nter) and SlitC-GFP (GFP in Cter). The fusion of Slit2N with GFP was done by cloning the Slit2 signal peptide, the eGFP and the Slit2N sequences in a pCAGEN backbone using the In-Fusion HD cloning Plus Kit (638909, Ozyme) with the following primers: 5′-GCA AAG AAT TCC TCG AGG ATA TCA TGC GCG GCG TTG GCT GG-3′ and 5′-TGC TCA CGT TAA CCG CCG GGC ACG CCT G-3′ for the signal peptide; 5′-CCC GGC GGT TAA CGT GAG CAA GGG CGA GG-3′ and 5′-AGC ACT GAC GCG TCT TGT ACA GCT CGT CCA TGC-3′ for eGFP; 5′-GTA CAA GAC GCG TCA GTG CTC TTG CTC GG-3′ and 5′-CTG AGG AGT GCG GCC GCG ATT TAA CGA GGG AGG ACC ATG GG-3′ for Slit2N. The CAG promoter was then replaced by the Hoxa1 promoter using In-Fusion with the following primers: 5′-GTG CCA CCT GGT CGA CGC TTC TTC TAG CGA TTA AAT C-3′ and 5′-AAC GCC GCG CAT GAT ATC CCC ACT AGT AAG CTT GGA GGT G-3′. The fusion of Slit2C with GFP was done by cloning the sequences into a pEGFP-N3 vector after the addition of KpnI and BamHI restriction sites using the following primers: 5′-CG GGTACC ACC AGC CCC TGT GAT AAT TTT G-3′ and 5′-CG GGATCC GGA CAC ACA CCT CGT ACA GC-3′. The Igκ Leader peptide signal was added for correct secretion by cloning the fusion fragment into a pSecTagB plasmid using KpnI and NotI digestion. The resulting sequence was cloned into a pCAGEN vector using XhoI and NotI digestion, and the In-Fusion HD Cloning Plus Kit (638909, Ozyme) with the following primers: 5′-CAA AGA ATT CCT CGA GAT GGA GAC AGA CAC ACT CCT GC-3′ and 5′-CTG AGG AGT GCG GCC GCT TAC TTG TAC AGC TCG TCC ATG CC-3′. The CAG promoter was replaced by the Hoxa1 promoter using SalI and XhoI digestion, and the In-Fusion HD Cloning Plus Kit with the following primers: 5′-GTG CCA CCT GGT CGA CGC TTC TTC TAG CGA TTA AAT CAA AG-3′ and 5′-CTG TCT CCA TCT CGA GCC CAC TAG TAA GCT TGG AGG TG-3′. Math-1-mbTomato construct was described previously (*Pignata et al., 2019*).

## DiI staining in spinal cord open-books

Spinal cords from E12.5 murine embryos were dissected and mounted as open-books prior to fixation in 4% paraformaldehyde (PFA) for 18 hr. DiI crystals (D3911, ThermoFisher) were inserted in the most dorsal part of one hemi-spinal cord for anterograde labeling of commissural tracts. Axon trajectories were analyzed 24 hr later with a spinning disk microscope (Olympus X80). For each DiI crystal, a range of phenotypes can be observed. Classes were made representing the percentage of DiI crystals showing the phenotype over the total number of observed DiI crystals. Classes were assessed independently, with percentage ranging from 0% to 100%.

## Immunofluorescence labeling

Cryosections from embryos collected at E12.5 were prepared and processed for staining as in *Charoy et al., 2012*. For some experiments, chick embryo sections and open-book spinal cords were blocked in 6% BSA (A7906, Sigma) and 0.5% Triton (T9284, Sigma) diluted in PBS for 5 hr at room temperature. Sections were incubated overnight at room temperature with anti-PlxnA1 antibody (gift from Y. Yoshida), anti-Robo3 antibody (1/100, R and D, AF3076); anti-L1CAM antibody (1:100, A439 Abcam 123990), anti-NgCAM antibody (1:50, 8D9, DSHB), anti-BEN (1:50, BEN, DSHB) or an anti-PC2 antibody (1:100, 3533, Abcam) in 1% BSA diluted in PBS. Alexa 488, Alexa 555 (1/500, Invitrogen) and Fluoroprobe 546 (1/400) were used as secondary antibodies.

## In ovo electroporation, open-book mounting and imaging

The neural tube of HH14/HH15 chick embryos was electroporated as described previously (*Delloye-Bourgeois et al., 2015*; *Pignata et al., 2019*). Plasmids were diluted in PBS with Fast Green (F7262, Sigma) at the following concentration:

- 1.5 mg/mL for dual-tagged Slit2 and dual-tagged Slit2 Δ. Lower concentration did not allow correct imaging with confocal microscopy.
- 1.34 mg/mL for Hoxa1-Slit2N-GFP and 0.94 mg/mL for Hoxa1-Slit2C-GFP. Both concentrations match the dual-tagged plasmid molarity in order to electroporate the same amount of plasmid.

- 0.5 mg/mL for Math1-mbTomato and 2 mg/mL for pHluo-PlxnA1 and pHluorin-PlxnA1$_{Y1815F}$ and 0,05 mg/mL for mbTomato and Hoxa1-GFP. These concentrations were selected according to our previous work (*Pignata et al., 2019*) in which we compared the outcome of different concentrations on the FP navigation and selected the conditions that gave the better compromise between receptor expression and ability of FP crossing. Thus, taking into account PlxnA1 endogenous expression, we used electroporation concentration that enabled the growth cones to enter the FP, those expressing too high levels being prevented from entry. The pHluo dependency was controlled by in vitro cell line transfection as in *Pignata et al., 2019*.
- In combinations with non-fluorescent PlxnA1, 1 mg/mL for pHluo-PlxnA1 was co-electroporated with 1 mg of his-PlxnA1$^{WT}$ and flag-PlxnA1$_{Y1815F}$.

The plasmid solution was injected into the lumen of the neural tube using picopritzer III (Micro Control Instrument Ltd., UK). Using electrodes (CUY611P7-4, Sonidel) three pulses (25V, 500 ms interpulse) were delivered by CUY-21 generator (Sonidell). Electroporated embryos were incubated at 38.5°C. In ovo electroporation of FP cells was done on HH17/18 chick embryos as described in *Wilson and Stoeckli, 2012*. Briefly, electrodes (CUY611P7-4, Sonidel) were placed at the thoracic level dorsally (cathode, negative electrode) and ventrally (anode, positive electrode), and three pulses (18V, 500 ms interpulse) were delivered by CUY21 electroporator (Sonidell). Embryos at HH25/HH26 were harvested in cold HBSS (14170–088, Gibco) and the spinal cords were dissected out. They were mounted in 0.5% agarose diluted in F12 medium on glass bottom dishes (P35G-1.5–14 C, MatTek). After agarose solidification, spinal cords were overlaid with 3 mL of F12 medium supplemented with 10% FCS (F7524; Sigma-Aldrich), 1% Penicillin/Streptomycin (Sigma-Aldrich) and 20 mM HEPES buffer (15630–049, ThermoFischer Scientific). For section imaging, embryos were then fixed for 2 hr at room temperature in 4% paraformaldehyde diluted in PBS. For open-book imaging, spinal cords were dissected and then fixed 45 min in 4% paraformaldehyde diluted in PBS. For vibratome sectioning, embryos were embedded in 3% low gelling agarose (A9414, Sigma) diluted in PBS. The embryos were then sectioned in 80 µm slices using Leica VT1000S vibratome.

## Mouse spinal cord electroporation and open-book mounting

E12 mice embryos were collected and fixed on a SYLGARD (Dow Corning) culture plate in HBSS medium (ThermoFisher) supplemented with Glucose 1M (Sigma-Aldrich). Injection of plasmids into the lumen of the neural tube was performed using picopritzer III (Micro Control Instrument Ltd., UK). Using electrodes (CUY611P7-4, Sonidel) three pulses (20V, 500 ms interpulse) were delivered by CUY-21 generator (Sonidell). Spinal cords were dissected from the embryos and cultured on Nucleopore Track-Etch membrane (Whatman) for 48 hr in Slice Culture Medium (*Polleux and Ghosh, 2002*).

## Imaging and data analysis

Live imaging was performed with an Olympus IX81 microscope equipped with a spinning disk (CSU-X1 5000 rpm, Yokogawa) and Okolab environmental chamber maintained at 37°C. Image was acquired with a 20× objective by EMCCD camera (iXon3 DU-885, Andor technology). Fifteen to thirty planes spaced of 0.5–3 µm were imaged for each open-book at 30 min interval for 10 hr approximately. To reduce exposure time and laser intensity, acquisitions were done using binning 2 × 2. Images were acquired using IQ3 software using multi-position and Z stack protocols. Z stack projections of the movies were analyzed in ImageJ software. The analysis of pHluo-flashes was performed from time-lapse acquisitions. In some experiments, the time interval was reduced for faster image acquisition. Time intervals of 3, 5, 8, and 12 min were tested. At 3 and 5 min, the tissues were rapidly damaged. We thus selected time interval of 8 min as the better compromise between time resolution and phototoxicity. Confocal imaging was performed with either an Olympus FV1000 with a 40× objective and zoom or a Leica TCS SP5 with a 63× objective. Deconvolution was done using the Huygens software. 3D surface reconstructions were done using the Imaris software.

## Atto647N staining and STED microscopy

Spinal cords were incubated at 38°C for 20 min with F12 medium supplemented with 5% FCS (F7524; Sigma-Aldrich), 20 mM HEPES buffer (15630–049, ThermoFischer Scientific), and 1/100 GFP-nanobodies Atto647N. They were then rinsed four times with the same medium free of GFP-nanobodies and fixed at room temperature for 2 hr with PBS supplemented with 4% paraformaldehyde

(PFA) and 1% BSA (A7638 Sigma-Aldrich). Open-books were observed with a STED microscope (TCS SP8, Leica). STED illumination of ATTO 647N was performed using a 633 nm pulsed laser providing excitation, and a pulsed bi-photon laser (Mai Tai; Spectra-Physics) turned to 765 nm and going through a 100 m optical fiber to enlarge pulse width (100ps) used for depletion. A doughnut-shaped laser beam was achieved through two lambda plates. Fluorescence light between 650 and 740 nm was collected using a photomultiplier, using a HCX PL-APO CS 100/1.40 NA oil objective and a pinhole open to one time the Airy disk (60 mm). Images were acquired using Leica microsystem software and a Z stack protocol. Usually 10–20 planes spaced of 0.5 µm where imaged for each growth cone. The growth cones were delineated and the intensity signal was calculated using ImageJ.

## Fluorescence recovery after photobleaching

FRAP experiments were conducted on spinal cord open-books electroporated with either pHluo-receptor and mbTomato, using a Leica DMI6000 (Leica Microsystems, Wetzlar, Germany) equipped with a confocal Scanner Unit CSU-X1 (Yokogawa Electric Corporation, Tokyo, Japan) and a scanner FRAP system, ILAS (Roper Scientific, Evry, France). Images were acquired in both green and red channels using a 63× objective and an Evolve EMCCD camera (Photometrics, Tucson, USA). Growth cones located in the FP were first monitored for 12 s each 3 s and then bleached using a 488 nm diode laser at full power. This resulted in an 85–95% loss of the signal (mean of 89%) at t = 0. Fluorescence recovery was then monitored for 730 s with acquisitions every 3 s for 30 s, then every 10 s for 100 s, and finally every 30 s for 600 s. The images were corrected for background noise, residual fluorescence right after the bleach was set to zero, and recovery curves were normalized to the fluorescence lost after the bleach. No other corrections were applied since unbleached growth cone fluorescence showed no significant decay during the acquisition period.

## Western blot

To observe Slit cleavage, N2a cells were seeded into 6-well plates ($2.5 \times 10^5$ cells per well). 24 hr later, cells were transfected using jetprime transfection reagent (114, Polyplus transfection). Four hours after starting the transfection, cell medium was changed and CMK (ALX-260–022, Enzo) was added to a final concentration of 100 µM if needed. Two days after transfection, CMK treatment was repeated. After 2 hr, cells were harvested. Whole cell extracts were isolated using RIPA buffer (NaCl 150 mM – Tris HCl pH 7, 35–50 mM – DOC 1% – N-P40 1% – $H_2O$) supplemented with protease inhibitor (04 693 116 001, Roche). Isolated protein concentration was determined using Bradford assay (500–0006, Bio-Rad).

Spinal cords were isolated from E12.5 embryos and dissected tissues were lysed in RIPA buffer supplemented with protease inhibitor. Samples were analyzed in western blot using anti-PlxnA1 antibody (Gift from Y Yoshoda), anti-GFP (1:1000, 11814460001, Sigma), anti-Tubulin (1:10000, T5168, Sigma), and anti-PC2 (1:100, 3533, Abcam). Western blot quantification was performed using Image Lab4.0 software (Bio-Rad).

## Commissural neuron cultures and collapse

Dorsal spinal cord tissue was dissected out from isolated spinal cord and dissociated. Neurons were grown on laminin-polylysine-coated coverslips in Neurobasal supplemented with B27, glutamine (Gibco), and Netrin-1 (R and D) medium for 24–48 hr, as in *Nawabi et al., 2010*. Immunolabeling was performed with anti-PlxnA1 antibody (gift from Y. Yoshida). Nuclei were stained with bisbenzi-mide (Promega) and actin with TRITC-phalloidin. GDNF was applied to the cultures as in *Charoy et al., 2012*. Collapse assays were performed as in *Delloye-Bourgeois et al., 2015*.

## Quantification and statistical analyses

Protein diffusion quantification: The background noise was measured in ImageJ and then subtracte. The electroporated zone was then divided into two different compartments along the dorso-ventral axis. The glial cells' apical feet and cellular bodies were included in the apical compartment. The axons path was divided in two, the most apical compartment being basal 1, the most basal compartment being basal 2. The mean intensity of Cerulean, Venus, or GFP was measured using ImageJ in

each compartment. The mean intensity was then normalized by the mean intensity in the apical compartment.

Pearson coefficient: The background noise was removed by measuring it, then subtracting it in ImageJ. The electroporated zone was then divided into three different compartments as mentioned previously. Each compartment was then analyzed using the JACoP plugin in ImageJ. An empirical threshold was used, and the Pearson's coefficient calculated.

All embryos that normally developed and express pHluo-vectors at the thoracic level were included in the analysis. Number of independent experiments, embryos, stacks and growth cones (n) are indicated in figures or legends. Analysis were done in blind for the quantification of phenotype in mouse embryos and the collapse assays. Sample size and statistical significances are represented in each figure and figure legend. For each set of data, normality was tested and Student's t-test or Mann–Whitney test was performed when the distribution was normal or not, respectively. Statistical tests were performed using Biosta-TGV (CNRS) and Prism six software.

## Acknowledgements

We thank Samir Merabet for helpful advices on the construction of fluorescent reporters, Julien Falk and Fréderic Moret for help with microscopy, Camilla Lucardini, Dennis Ressnikoff, and Bruno Chapuis from the CIQLE platform of Lyon for advices on microscopy and deconvolution, Esther Stoeckli for Math1 and HoxA1 promoter constructs. STED microscopy was done in the Bordeaux Imaging Center (BIC), CNRS-INSERM-Bordeaux University, member of the national infrastructure France Biolmaging supported by the French National Research Agency (ANR-10-INBS-04). We thank C Poujol and M Mondin (BIC) for advice, M Sainlos for sharing the anti-GFP nanobody, and B Tessier and S Benquet for technical assistance. This work was conducted within the frame of the LabEx CORTEX and DEVWECAN of Universite′ de Lyon, within the program "Investissements d'Avenir" (ANR-11-IDEX-0007) operated by the French National Research Agency (ANR). This study was supported by an ANR funding to VC and OT, the Association Française contre les Myopathies (AFM), the Fondation pour la Recherche Médicale (FRM) to VC, the Fondation Bettencourt-Schueller to VC, and Conseil Régional d'Aquitaine (Neurocampus funds).

## Additional information

### Funding

| Funder | Grant reference number | Author |
|---|---|---|
| Agence Nationale de la Recherche | ANR-11-IDEX-0007 | Valerie Castellani Olivier Thoumine |
| Fondation pour la Recherche Médicale | Label team | Valerie Castellani |
| Fondation Bettencourt Schueller | | Valerie Castellani |
| AFM-Téléthon | | Valerie Castellani Hugo Ducuing |

The funders had no role in study design, data collection and interpretation, or the decision to submit the work for publication.

### Author contributions

Hugo Ducuing, Thibault Gardette, Formal analysis, Investigation, Methodology, Writing - original draft; Aurora Pignata, Karine Kindbeiter, Muriel Bozon, Céline Delloye-Bourgeois, Formal analysis, Methodology; Olivier Thoumine, Methodology; Servane Tauszig-Delamasure, Formal analysis, Supervision, Methodology, Writing - original draft; Valerie Castellani, Conceptualization, Formal analysis, Supervision, Funding acquisition, Methodology, Writing - original draft

## Author ORCIDs

Hugo Ducuing https://orcid.org/0000-0001-8227-7924
Olivier Thoumine http://orcid.org/0000-0002-8041-1349
Servane Tauszig-Delamasure http://orcid.org/0000-0003-4926-0199
Valerie Castellani https://orcid.org/0000-0001-9623-9312

## Decision letter and Author response

Decision letter https://doi.org/10.7554/eLife.63205.sa1
Author response https://doi.org/10.7554/eLife.63205.sa2

## Additional files

### Supplementary files

• Transparent reporting form

### Data availability

All data generated or analysed during this study are included in the manuscript and supporting files. Source data files have been provided for figures.

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
