## [Decision Letter]

**Acceptance summary:**

The results from this study provide a novel view on how spinal cord commissural axons -a commonly used paradigm to study mechanism of axon guidance- grow across the floor plate, a glial-like structure positioned at the ventral spinal cord midline. The study shows that a well-known axon guidance cue, SlitC, deposited on very elaborated basal end-feet of floor plate cells, constantly limits the exploration of the commissural growth cones expressing its PlexinA1 receptor. Thus, this receptor/ligand pair imposes order to axons as they cross to the contralateral side of the spinal cord. This observation rectifies the current view whereby Slits were thought to provide a repulsive signal that prevented axons to turn back after crossing.

**Decision letter after peer review:**

Thank you for submitting your article "SlitC-PlexinA1 mediates iterative inhibition for orderly passage of spinal commissural axons through the floor plate" for consideration by *eLife*. Your article has been reviewed by three peer reviewers, and the evaluation has been overseen by a Reviewing Editor and Marianne Bronner as the Senior Editor. The reviewers have opted to remain anonymous.

The reviewers have discussed the reviews with one another and the Reviewing Editor has drafted this decision to help you prepare a revised submission.

Summary:

This is an interesting and elegant study that takes advantage of time-lapse imaging and interference with the Slit/Plexin interaction to obtain a dynamic view of commissural axon navigation through the floor plate midline, a model that has been extensively exploited to understand core mechanisms of axon guidance. In agreement with previous studies, the authors show that commissural axons make close contacts with floor plate cells as they navigate through it. More importantly they provide an elaborate analysis of floor plate basal processes, their changes and their differential distribution of cleaved Slit fragments, showing how these prevent commissural growth cones from exploring the environment and direct axonal trajectories forward until they exit the floor plate. The authors further show that phosphorylation of PlexinA1 at Y1815 is essential for restricting exploratory growth cone behaviors.

Overall, the study is solid and provides innovative ideas on growth cone navigation and receptor/ligand interactions during axon guidance with a well performed analysis. Nevertheless, there are a number of issues that need to be addressed.

Essential revisions:

1) The study heavy relies on the expression of a mutant form of Plexin (Plexin^Y1815F^). However, the functioning of the mutation is insufficiently described and its effects confusing. The authors need to clarify why and how "the Y1815 mutation alters PlxnA1 cell surface pattern". Does the mutant form when overexpressed act dominantly, altering the distribution of endogenous protein? Do the PlexinA1-Y1815F/ PlexinA1-wt mice have a phenotype? Have the authors tested a PlexinA1-Y1815 Alanin mutation as a counterpart to PlexinA1-Y1815F? The effect of the Plexin mutant form is particularly important in the electroporation experiments in chick embryos where two copies of endogenous PlexinA1 are present. How do the authors explain that these electroporated chick commissural neurons show a phenotype similar to the mutant mice that only express the PlexinA1^Y1815F/Y1815F^ form? It would be also helpful if the authors discuss potential problems in relying in their interpretations mostly on this paradigm as the mutant form could disrupt the dynamic/temporal control of receptor trafficking or signaling in general.

2) A phenotype similar to that described here was reported by Hutson and Chien, 2002, analyzing the zebrafish astray mutants. In that study, the authors proposed a role for Slit-Robo signaling in preventing and correcting pathfinding errors. These data need to be mentioned and discussed. Is the "straight growth" and dampening of exploration proposed here, similar to the restriction of error and error correction model proposed by Hutson and Chien?. Could PlxnA1-SliC signaling be playing a similar role at the floorplate to Robo-Slit signaling in the visual system? Furthermore, Slit signaling has been proposed to play a barrier function that is important for channeling axons along their correct path. How might this relate to the model being proposed here? How "radically different" is the mode of guidance being proposed here from that described already in other parts of the CNS?

3) Throughout the manuscript, the authors refer to an "unexpectedly" complex mesh of ramified complex FP glial basal processes". However, there have been several previous studies of glia at the floor plate and other regions of the CNS midline using immunohistochemistry and electron microscopy. These studies have demonstrated that midline glia from a dense palisade and are highly branched and that complex interactions occur between the glial cells and axons traversing the midline (e.g. Marcus et al., 1995; Campbell and Peterson, 1993). The finding reported here that the glial cell basal processes from a complex mesh of ramified processes is therefore not unexpected. While the 3D reconstructions are beautiful and add some detail to our understanding of the interactions between axons and glial cells at the CNS midline, it needs to be made clearer, particularly in the Results sections, that these analyses largely confirm previous work and what the specific advances made here are. In this respect, the complex end feet processes of midline glia might have important implications regarding ligand presentation. It would seem interesting to test conditions where end feet morphologies of the glia cells are lost. In a number of different studies people have used expression of dominant RhoA or cdc42 mutant proteins to prevent such cellular process formation. Do the authors consider it feasible to use here?

---

## [Author Response]

Essential revisions:1) The study heavy relies on the expression of a mutant form of Plexin (Plexin^Y1815F^). However, the functioning of the mutation is insufficiently described and its effects confusing. The authors need to clarify why and how "the Y1815 mutation alters PlxnA1 cell surface pattern". Does the mutant form when overexpressed act dominantly, altering the distribution of endogenous protein? Do the PlexinA1-Y1815F/ PlexinA1-wt mice have a phenotype? Have the authors tested a PlexinA1-Y1815 Alanin mutation as a counterpart to PlexinA1-Y1815F? The effect of the Plexin mutant form is particularly important in the electroporation experiments in chick embryos where two copies of endogenous PlexinA1 are present. How do the authors explain that these electroporated chick commissural neurons show a phenotype similar to the mutant mice that only express the PlexinA1^Y1815F/Y1815F^ form? It would be also helpful if the authors discuss potential problems in relying in their interpretations mostly on this paradigm as the mutant form could disrupt the dynamic/temporal control of receptor trafficking or signaling in general.

We fully agree that these are interesting questions for which we do have partial answers.

PlxnA1^Y1815F^ electroporation in the chicken embryo can be seen as a mimic of heterozygosity, with growth cones expressing both the mutated and the endogenous WT receptor. Their respective concentration might be in favor of the mutated receptor, due to the over-expression condition. We focused our analysis to the growth cones that had sorted PlxnA1^Y1815F^ receptor at their cell surface, even though the endogenous receptor might also be present. Several mechanisms can be considered:

– First, growth cones preferentially engaging PlxnA1^Y1815F^ interactions with SlitC for various reasons (availability, stability, affinity, etc.) behave abnormally.

– Second, competition for cell surface sorting due to increased PlexinA1 receptor expression could lead to depletion of WT receptor in some growth cones.

– Third, PlxnA1^Y1815F^ could titrate PlxnA1^WT^ receptor in contexts where both are expressed, by forming dysfunctional complexes.

We can discuss these hypotheses in light of the phenotypes observed in our mouse model. We do see recrossing phenotypes in the PlxnA1^Y1815F/+^ heterozygous embryos (Figure 1C). They look similar to those observed in the PlexinA1^Y1815F/Y1815F^ embryos. This indicates that the mutated PlxnA1 receptor can alter SlitC signaling in presence of the WT one. In contrast, PlxnA1^+/-^ embryos do not exhibit recrossing phenotypes (Delloye-Bourgeois et al., 2015), which thus suggests that one copy of PlxnA1 might be sufficient to ensure SlitC-PlxnA1 signaling. This weakens the relevance of the model of depletion by competition for sorting; leaving more plausible hypotheses 1 and 3. Addressing these models is difficult because we have no ways of distinguishing PlxnA1^WT^ and PlexinA1^Y1815F^ receptors in the mouse model as they are both recognized by anti-PlexinA1 antibodies, in western blot and in immunohistochemistry. In the chicken, PlxnA1 antibodies that we tested recognized both chick and mouse PlxnA1, and we thus cannot specifically assess the expression of the endogenous receptor. Nevertheless, we attempted to get some more information using the chicken embryo model. We made an in vivo competition assay by diluting the dose of electroporated pHLuo-PlxnA1^WT^ with either PlxnA1^WT^ or PlxnA1^Y1815F^ (using non-fluorescent receptors forms) and we compared the cell surface distribution of pHluo-PlxnA1^WT^ in commissural axons navigating the FP. We observed a similar enrichment of the pHluo-PlxnA1^WT^ in the growth cone compartment, compared to the adjacent axon segments. This indicates that co-expression with PlxnA1^Y1815F^ does not affect the compartmentalization of pHluo-PlxnA1^WT^. Nevertheless, it did not exclude the possibility that PlxnA1^Y1815F^ titrates PlxnA1^WT^ if both receptors dimerize, which would result in depleting the growth cones of the PlxnA1 receptors that are normally able to transduce SlitC signal. We thus quantified the pHluo signal in the growth cones and found no significant differences in the presence or absence of the mutated receptor. Thus, these experiments support a model whereby PlxnA1^Y1815^ has a direct contribution to the altered growth cone behaviors, even in the presence of the PlxnA1^WT^ receptor. These data are presented in Figure 3—figure supplement 1, subsection “The temporal and spatial pattern of PlxnA1 membrane insertion during commissural axon navigation is impacted by the Y1815 mutation” and discussed in the subsection “Tyrosine mutation alters the membrane mobility of PlxnA1 receptor”.

We did not engineer a PlxnA1 with Y1815 Alanin mutation. We thought that the fact that the Y1815F mutation only altered SlitC but not Sema3B responses was a good internal control of the substitution.

2) A phenotype similar to that described here was reported by Hutson and Chien, 2002, analyzing the zebrafish astray mutants. In that study, the authors proposed a role for Slit-Robo signaling in preventing and correcting pathfinding errors. These data need to be mentioned and discussed. Is the "straight growth" and dampening of exploration proposed here, similar to the restriction of error and error correction model proposed by Hutson and Chien?. Could PlxnA1-SliC signaling be playing a similar role at the floorplate to Robo-Slit signaling in the visual system? Furthermore, Slit signaling has been proposed to play a barrier function that is important for channeling axons along their correct path. How might this relate to the model being proposed here? How "radically different" is the mode of guidance being proposed here from that described already in other parts of the CNS?

We thank the reviewers for this comment and indeed we apologize for having missed the cited study. After reading, we fully agree that our findings must be discussed in light of this study, which reported in the context of retinal axon navigation observations than echo ours. Astray retinal growth cones are observed to be more complex and less streamlined than WT ones, deviating their trajectory from the normal path, already prior to reaching the midline, which correlates with Slit expression patterns framing the navigation path. Robo2-Slit might thus maintain retinal growth cones straight on their way towards and after midline crossing, as we propose PlxnA1-SlitC does so in the context of the floor plate navigation.

We have now included a paragraph discussing this work in the subsection “Is PlxnA1-SlitC signaling the only player of the guidance program ensuring straight trajectory across the FP?”.

3) Throughout the manuscript, the authors refer to an "unexpectedly" complex mesh of ramified complex FP glial basal processes". However, there have been several previous studies of glia at the floor plate and other regions of the CNS midline using immunohistochemistry and electron microscopy. These studies have demonstrated that midline glia from a dense palisade and are highly branched and that complex interactions occur between the glial cells and axons traversing the midline (e.g. Marcus et al., 1995; Campbell and Peterson, 1993). The finding reported here that the glial cell basal processes from a complex mesh of ramified processes is therefore not unexpected. While the 3D reconstructions are beautiful and add some detail to our understanding of the interactions between axons and glial cells at the CNS midline, it needs to be made clearer, particularly in the Results sections, that these analyses largely confirm previous work and what the specific advances made here are. In this respect, the complex end feet processes of midline glia might have important implications regarding ligand presentation. It would seem interesting to test conditions where end feet morphologies of the glia cells are lost. In a number of different studies people have used expression of dominant RhoA or cdc42 mutant proteins to prevent such cellular process formation. Do the authors consider it feasible to use here?

We fully agree that previous studies reported that commissural axons navigate within the network of basal processes of the floor plate glia cells. Our claim of “unexpectedly complex mesh of ramified process” was in fact referring to the multiple ramifications of the basal process, the particular asymmetric morphologies these cells have, and how they are arranged in 3D in the tissue, which were not reported so far. We modified the manuscript according to the suggestions, adding sentences to make it clearer what our observations bring to the known background, Results section and Discussion. We also extended the literature in this paragraph, citing previous studies reporting glia cell specializations and their contribution to axon guidance. We toned down the novelty of the findings in the manuscript and Abstract, removing “novel” at several places.

We fully agree that manipulating the morphology of the FP cells would be very interesting to examine whether it modifies the ligands distribution and alters commissural axon navigation. The consequences of Rho ablation in the developing spinal cord have been examined using a conditional line with specific ablation of RhoA in neural progenitors. The study showed that it essentially altered the apical polarity of the progenitors, disrupting adherens junctions. In contrast, the basal polarity was reported to be unaffected (Herzog et al., J. Neurosci 2011). In another study, a dominant negative RhoA was electroporated in the chick neuroepithelium. The major reported consequence was an alteration of the division angle of neural progenitors at the apical side (Roszko et al., Dev Biol, 2006). It seems difficult from this literature to predict that altering RhoA would result in a disruption of FP glia basal processes, and we feel that addressing this point in the context of revisions would take a long time because of the technical challenges it represents.

Beyond the experimental difficulty to reach the objective, the observations would be difficult to interpret because altering the basal process would not only impact the distribution of the ligands but would also change the physical properties of the navigation path that may also account for proper navigation.